# Discrete Latent Features Ablate Adversarial Attack: A Robust Prompt Tuning Framework for VLMs

**Yang Chen**[1*]**, Yanbin Wei**[1,2*]**, James Kwok,**[2]**, Yu Zhang**[1†]

[1]Department of Computer Science and Engineering, Southern University of Science and Technology
[2]Department of Computer Science and Engineering, Hong Kong University of Science and Technology
`cheny2023@mail.sustech.edu.cn`
`{yanbin.ust, yu.zhang.ust}@gmail.com`
`jamesk@cse.ust.hk`

## Abstract

While adversarial fine-tuning can enhance the robustness of vision-language models (VLMs), such approaches are computationally expensive. Adversarial prompt tuning has emerged as a practical alternative. However, existing methods are limited by their reliance on vulnerable continuous image features. To mitigate the vulnerability in the feature representation, we propose **DEFEAT** (**D**iscrete Lat**E**nt **F**eatur**E** based **A**dversarial **T**raining), a robust prompt tuning framework for VLMs. Specifically, the DEFEAT method introduces a perturbation discrete shield module that reconstructs discrete latent features and designs a logits fusion strategy, substantially reducing the discrepancy between clean and adversarial image representations. Moreover, the DEFEAT method integrates prompt tuning with adversarial training while applying regularization from learnable prompts to hand-crafted prompts, further enhancing the adversarial robustness. Extensive experiments across 15 datasets validate the effectiveness of the proposed DEFEAT method among existing adversarial prompt tuning methods. The official code is available at `https://github.com/cheny02/DEFEAT-ICLR2026`.

## 1 Introduction

In recent years, vision-language models (VLMs), particularly CLIP (Radford et al., 2021) and its successors (Sun et al., 2023; 2024; Zhang et al., 2024a; Li et al., 2022; Jia et al., 2021), have demonstrated remarkable performance across various visual tasks. As these models are deployed for real-world applications, it is crucial to address their vulnerabilities. Recent studies (Touvron et al., 2023; Zou et al., 2023; Schlarmann & Hein, 2023; Zhao et al., 2023; Carlini et al., 2023) have shown that VLMs are highly susceptible to adversarial examples (Szegedy et al., 2013), which can lead to significant security issues by possessing imperceptible perturbations.

Adversarial training (Madry et al., 2018) is widely regarded as the most effective defense against such attacks. While many studies (Mao et al., 2023; Schlarmann et al., 2024; Hossain & Imteaj, 2024; Gong et al., 2025) have improved the robustness of CLIP-like models by adversarially fine-tuning the entire model, this approach is impractical for large-scale deployment due to the immense parameter counts of modern VLMs. To improve efficiency, Parameter-Efficient Fine-Tuning (PEFT) techniques (Ding et al., 2023) have emerged as a promising solution to enable the adaptation of large models by tuning only a small subset of parameters. Building on this, Li et al.

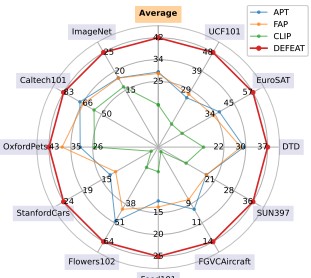

Figure 1: The harmonic mean accuracy of the robustness ($\epsilon = 4/255$) and accuracy of baselines and the proposed DEFEAT method under the adversarial few-shot classification setting.

---

[*]Equal contribution.
[†]Corresponding author.

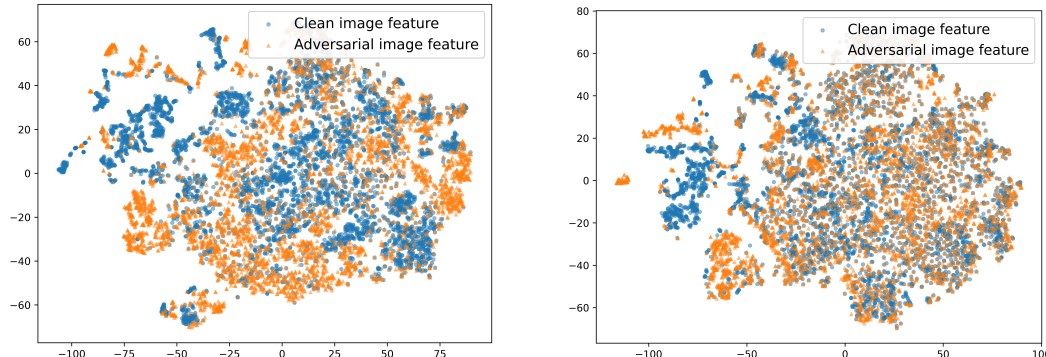

Figure 2: The left and right images illustrate the latent image features without and with the feature discretization, respectively.

(2024); Zhou et al. (2024); Zhang et al. (2024b) have integrated adversarial training with prompt tuning (Zhou et al., 2022b), a PEFT method highly effective for VLMs, to efficiently enhance the robustness on downstream tasks.

Among those existing methods, most of them rely on continuous image features for adversarial defense and mainly focus on how to train robust models. In contrast, we mainly focus on the feature representation, a different direction. After exploring this direction, we have a finding that *discretizing the latent image feature could effectively mitigate adversarial attacks*. To see that, as illustrated in Figure 2, latent features processed with a discretization exhibit a significantly smaller shift between clean and adversarial examples than continuous image features, where the experimental settings are introduced in Section 3.2. This indicates that the latent feature discretization could reduce the impact of feature shifts induced by adversarial attacks.

Based on this finding, we propose a **D**iscrete Lat**E**nt **F**eatur**E** based **A**dversarial **T**raining (**DEFEAT**) method, a robust prompt tuning framework for VLMs. DEFEAT designs a perturbation discrete shield (PerturbShield) module that mitigates adversarial attacks through grid-based discrete feature reconstruction and semantic alignment projection. To optimize the trade-off between the robustness and accuracy, the DEFEAT method uses a logits fusion strategy that combines predictions from PerturbShield and original image features. Additionally, DEFEAT applies regularization to the learnable prompt using hand-crafted prompts and demonstrates that this regularizer enhances the adversarial robustness. Extensive experiments across 15 datasets demonstrate that the proposed DEFEAT method achieves state-of-the-art performance compared to existing methods and offers an improved trade-off between the robustness and performance (see Figure 1). Specifically, the proposed DEFEAT method achieves an average improvement of 13.76% in terms of the harmonic mean of the robustness and accuracy compared to the previous state-of-the-art method under the adversarial few-shot classification setting.

Our contributions are three-fold. (i) To the best of our knowledge, we are the first to use latent feature discretization to mitigate adversarial attacks. (ii) We analyze the mitigating effect of feature discretization on visual adversarial perturbation and propose the DEFEAT method as a robust prompt tuning framework for VLMs; (iii) Extensive experiments on 15 datasets demonstrate the effectiveness of the proposed DEFEAT method under various settings, including adversarial few-shot classification, adversarial domain generalization, and adversarial cross-dataset generalization.

## 2 RELATED WORK

**CLIP-based VLMs.** VLMs have significantly boosted cognitive capabilities by merging visual and textual modalities, excelling in real-world vision tasks (Liu et al., 2023; Zhu et al., 2024). The introduction of CLIP (Radford et al., 2021), trained on about 400 million image-text pairs, was particularly transformative, establishing a new paradigm for vision-language representation learning. Numerous subsequent works have followed this paradigm, proposing a broad family of CLIP-like models, including ALIGN (Jia et al., 2021), EVA-CLIP (Sun et al., 2023), OpenCLIP (Ilharco et al.,

2021), and LongCLIP (Zhang et al., 2024a). Given the trailblazing role and widespread adoption of this architecture, we focus our work on CLIP as the representative model to investigate robust prompt tuning within this foundational VLM paradigm.

**Prompt tuning for VLMs.** Prompt tuning (Zhou et al., 2022b; Khattak et al., 2023a; Chen et al., 2025; Zhou et al., 2022a; Khattak et al., 2023b) has emerged as a popular lightweight model adaptation technique, particularly in VLMs. Unlike conventional fine-tuning approaches that involve updating all parameters of a pre-trained model, prompt tuning focuses on optimizing only the prompt representations, which significantly reduces computational overhead. CoOp (Zhou et al., 2022b) introduces an efficient adaptation strategy by optimizing learnable prompt vectors that replace manually crafted textual prompts (e.g., "a photo of a panda") in the textual branch of CLIP, marking a significant milestone in the vision-language alignment. However, CoOp tends to overfit base classes with limited generalization ability. To address this, several studies (Zhu et al., 2023; Yao et al., 2023; Chen et al., 2024; Khattak et al., 2023b) use the knowledge captured in pre-trained CLIP through hand-crafted prompts to enhance the generalization of learnable prompts. In this paper, our study builds on prompt tuning of CoOp to explore robust prompt tuning for VLMs.

**Adversarial training of VLMs.** Adversarial attacks (Madry et al., 2018; Goodfellow et al., 2014) fool models by adding imperceptible perturbations to input images. Several studies (Touvron et al., 2023; Zou et al., 2023; Schlarmann & Hein, 2023; Zhao et al., 2023; Carlini et al., 2023) have shown that VLMs are highly vulnerable to adversarial attacks. Among various defense strategies, adversarial training (Madry et al., 2018; Zhang et al., 2019; Goodfellow et al., 2014; Zhang et al., 2020) is one of the most effective defense methods against such attacks. It strengthens the model robustness by incorporating adversarial examples into the training data and optimizing the model to classify both clean and adversarial examples correctly. Building on this idea, TeCoA (Mao et al., 2023) enhances adversarial robustness in CLIP through supervised adversarial fine-tuning. FARE (Schlarmann et al., 2024) introduces an unsupervised adversarial fine-tuning framework to obtain a robust CLIP, which preserves nominal performance while transferring robustness to downstream tasks. However, fine-tuning the entire model is not so economical. Alternatively, AdvPT (Zhang et al., 2024b) and APT (Li et al., 2024) combine prompt tuning with adversarial training, introducing adversarial prompt tuning for CLIP to enhance adversarial robustness through a PEFT way. Building on multi-modal prompt tuning (Khattak et al., 2023a), FAP (Zhou et al., 2024) improves the consistency of multimodal features and encourages differentiation in unimodal features between clean and adversarial examples, thereby enhancing adversarial robustness. Unlike previous adversarial prompt tuning methods that focus on learning robust prompts, the proposed DEFEAT method integrates VQ-VAE to mitigate the impact of adversarial attacks, achieving a better trade-off between robustness and accuracy.

## 3 METHODOLOGY

In this section, we begin with a brief overview of CLIP, adversarial attacks and training in CLIP, and VQ-VAE. Then we analyze the impact of VQ-VAE on visual adversarial perturbation. After that, we introduce the proposed DEFEAT in detail.

### 3.1 PRELIMINARIES

**CLIP.** CLIP facilitates zero-shot image classification by evaluating the similarity between visual and textual embeddings. It comprises a visual encoder and a text encoder. Let $\mathbf{I}$ represent the class feature extracted by CLIP's visual encoder for an image, which serves as the image representation and is generated by a class token. $\mathbf{t} = \{\mathbf{t}_c\}_{c=1}^C$ denotes a collection of textual embeddings produced by the text encoder for textual prompts, where each $\mathbf{t}_c$ corresponds to the textual embedding for class $c$, and $C$ is the number of classes. Those prompts typically follow the format "a photo of a [CLS]", where "[CLS]" is a placeholder for class tokens that are specific class names like "panda", "dog", and "bird". Based on those embeddings, the probability of classifying an image to belong to a class can be calculated as $p^{\mathrm{clip}}(y|\mathbf{x}) = \frac{\exp\left(f_y(\mathbf{x})/\tau\right)}{\sum_{c=1}^C \exp\left(f_c(\mathbf{x})/\tau\right)}$, where $\tau$ is the temperature parameter, $f_c(\mathbf{x}) = \cos(\mathbf{t}_c, \mathbf{I})$, and $\cos(\cdot, \cdot)$ denotes the cosine similarity.

**Adversarial attack and training in CLIP.** An adversarial attack (Madry et al., 2018) fools the model by learning an imperceptible perturbation $\delta$ added to a clean example $\mathbf{x}$, generating an adversarial example $\mathbf{x}_{\mathrm{a}}$. Specifically, in CLIP, the adversarial example $\mathbf{x}_{\mathrm{a}}$ is optimized by maximizing the

image-to-text contrastive loss (Mao et al., 2023; Li et al., 2024; Zhou et al., 2024) as

$$\mathbf{x}_{\mathrm{a}} = \arg\max_{\mathbf{x}_{\mathrm{a}}} \mathcal{L}(\mathbf{x}_{\mathrm{a}}, \mathbf{t}, \mathbf{y}), \ \text{s.t.} \ \mathbf{x}_{\mathrm{a}} = \mathbf{x} + \delta, \ \|\delta\|_p \leqslant \epsilon, \tag{1}$$

where $\mathcal{L}(\mathbf{x}_{\mathrm{a}}, \mathbf{t}, \mathbf{y}) = -\sum_{i=1}^{C} y_i \log p^{\mathrm{clip}}(y_i|\mathbf{x}_{\mathrm{a}})$, label $\mathbf{y}$ is a one-hot vector with $y_i$ equal to 1 when class $i$ is the ground-truth label for $\mathbf{x}$ and 0 otherwise, $\epsilon$ denotes the perturbation size, and $\|\cdot\|_p$ denotes the $\ell_p$ norm of a vector. Here we focus on the $\ell_\infty$ threat model (i.e., $p = \infty$).

After generating adversarial examples, Mao et al. (2023) employs adversarial training to optimize the parameters $\theta$ of CLIP's vision encoder, thereby developing a robust CLIP. The objective of adversarial training is formulated as $\theta = \arg\min_\theta \mathcal{L}(\mathbf{x_a}, \mathbf{t}, \mathbf{y})$.

**VQ-VAE.** VQ-VAE is a generative model based on discrete latent representations, with its core mechanism to convert continuous representations into discrete codes. Specifically, VQ-VAE consists of an encoder $E$, a decoder $D$, and a codebook $\mathbf{e}$. The encoder maps input data $\mathbf{x} \in \mathbb{R}^{H \times W \times C}$ into a continuous latent vector $\mathbf{z}_e = E(\mathbf{x}) \in \mathbb{R}^{n \times d}$, where $d$ is the dimensionality of the latent embedding, $n$ is the number of latent positions after encoding. The learnable codebook is defined as $\mathbf{e} = \{\mathbf{e}_k\}_{k=1}^{K}$, where $K$ is the size of the discrete latent space. The latent vector $\mathbf{z}_e$ is then quantized into a discrete latent representation, $\mathbf{z}_q \in \mathbb{R}^{n \times d}$. This is achieved by replacing each vector in $\mathbf{z}_e$ with its nearest neighbor from the codebook:

$$\mathbf{z}_q^{(i)} = \arg\min_{\mathbf{e}_k \in \mathbf{e}} \|\mathbf{z}_e^{(i)} - \mathbf{e}_k\|_2 \ , \forall i \in \{1, ..., n\}, \tag{2}$$

where $\mathbf{z}_q^{(i)}$ and $\mathbf{z}_e^{(i)}$ are the vectors at the $i$-th position of $\mathbf{z}_q$ and $\mathbf{z}_e$, respectively, and $\|\cdot\|_2$ denotes the $\ell_2$ norm. The quantized discrete representation $\mathbf{z}_q$ is then used to reconstruct the input by the decoder as $\hat{\mathbf{x}} = D(\mathbf{z}_q)$. Since there is no real gradient defined for Eq. (2), VQ-VAE copies gradients from the decoder input $\mathbf{z}_q$ to the encoder output $\mathbf{z}_e$. The entire training objective in VQ-VAE is formulated as

$$\mathcal{L}_{\mathrm{VQ\text{-}VAE}}(\mathbf{x}) = \alpha \underbrace{\|\mathbf{x} - \hat{\mathbf{x}}\|_2^2}_{\text{reconstruction loss}} + \beta \underbrace{\|\mathrm{sg}[\mathbf{z}_e] - \mathbf{z}_q\|_2^2}_{\text{codebook loss}} + \gamma \underbrace{\|\mathbf{z}_e - \mathrm{sg}[\mathbf{z}_q]\|_2^2}_{\text{commitment loss}}, \tag{3}$$

where $\mathrm{sg}[\cdot]$ is the stop gradient operator. By following the original implementation (van den Oord et al., 2017), $\gamma$ is set to be $0.25\beta$. Since VQ-VAE is an end-to-end, trainable, and commonly used discretization model, we use it to discretize and reconstruct latent image features.

## 3.2 MITIGATING EFFECT OF VQ-VAE ON VISUAL ADVERSARIAL PERTURBATION

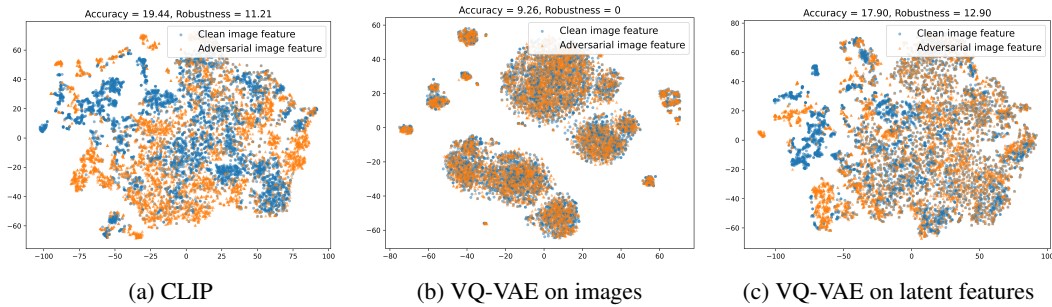

(a) CLIP      (b) VQ-VAE on images      (c) VQ-VAE on latent features

Figure 3: Visualization of clean and adversarial examples on the EuroSAT dataset. The VQ-VAE is trained with 16-shot samples per class under the adversarial few-shot classification setting.

In this section, we explore how VQ-VAE mitigates visual adversarial perturbations. We first visualize the clean and adversarial image features output by the CLIP's image encoder in Figure 3a. We observe that imperceptible perturbations ($\epsilon = 4/255$) cause a significant shift of latent features between clean and adversarial examples, leading to a sharp decline in classification accuracy for adversarial samples (i.e., from 19.44% to 11.21%).

Inspired by the quantization process of VQ-VAE, we hypothesize that the complex discretization process could mitigate adversarial attacks by reducing the distribution shift between clean and

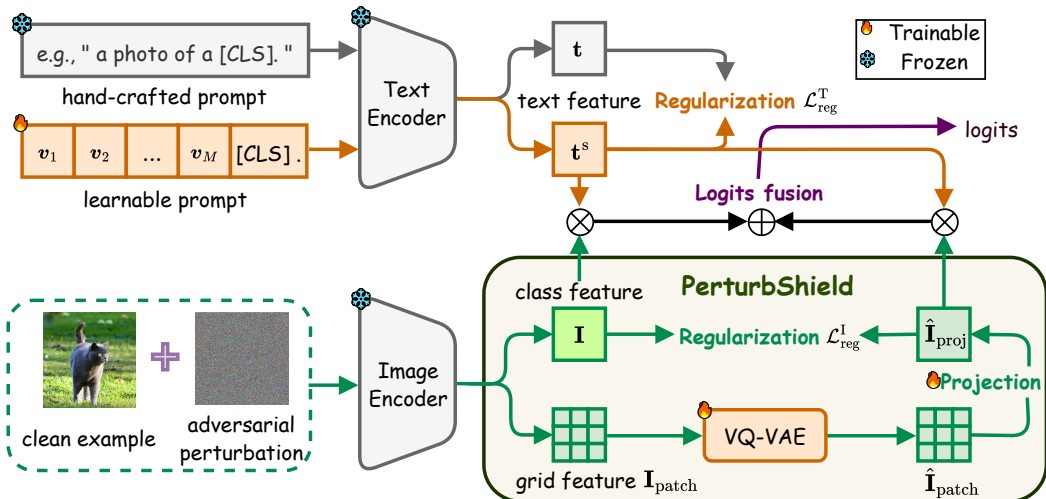

Figure 4: Overview of the proposed DEFEAT method. Note that both image and text encoders remain frozen, with only the learnable prompt and the perturbation discrete shield module being trained. In our experiments, we exclusively use adversarial examples for training.

adversarial examples. To validate this hypothesis, we integrate VQ-VAE as a defense module within the VLM framework against adversarial attacks. Firstly, we attempt to use VQ-VAE to reconstruct input images as did in (Mao et al., 2022). As shown in Figure 3b, the shift of latent features between clean and adversarial examples is significantly reduced, demonstrating that the discretization process can suppress adversarial perturbations. However, pixel-level reconstruction with VQ-VAE introduces significant information loss and structural distortions that compromise semantic content preservation when training data is limited. To see that, we can find that the feature distributions in Figure 3b differ greatly from the original image feature distributions shown in Figure 3a. The poor performance (i.e., Accuracy=9.26%, Robustness=0%) could verify this and also make this approach impractical.

Alternatively, we use VQ-VAE to reconstruct the latent feature output by ViT. As shown in Figure 3c, we find that the distribution shift of latent features between adversarial and clean examples is significantly reduced. Moreover, the distribution of latent features after VQ-VAE (shown in Figure 3c) is more similar to the feature distribution by the CLIP (shown in Figure 3a). This suggests that reconstructing latent features can also effectively suppress adversarial perturbations and achieve better performance (i.e., Accuracy=17.90%, Robustness=12.90%). One possible reason is that when adversarial perturbations are introduced into clean image features, the process of finding the nearest codes often makes both clean and adversarially perturbed features be assigned to the same discrete representation (i.e., codes). Hence, the discretization of VQ-VAE effectively neutralizes the perturbations within its discrete representation space, thus hindering the updates of adversarial perturbations and diminishing the distributional differences between clean and adversarial examples.

### 3.3 DEFEAT: Discrete Latent Feature based Adversarial Training

Motivated by the above observation, we propose the DEFEAT method for CLIP-based VLMs. Besides a frozen image encoder and a frozen text encoder in CLIP, DEFEAT consists of a learnable prompt and a perturbation discrete shield module. The architecture of DEFEAT is illustrated in Figure 4.

**Prompt tuning for text encoder.** We introduce learnable prompt vectors $\mathbf{v} = \{\mathbf{v}_1, \mathbf{v}_2, \ldots, \mathbf{v}_M\}$ to replace the textual prompt (e.g., "a photo of a"), where each $\mathbf{v}_i$ has the same dimension as the word embeddings. The learnable prompt is subsequently constructed by concatenating $\mathbf{v}$ with the class token. Formally, the learnable prompts are formed as $\{\mathbf{v}_1, \mathbf{v}_2, \ldots, \mathbf{v}_M, [\text{CLS}]\}$. Let $\mathbf{t}^s = \{\mathbf{t}^s_c\}_{c=1}^C$ be a set of textual embeddings generated by the text encoder for the learnable prompts, where each $\mathbf{t}^s_c$ corresponds to the textual embedding of learnable prompt for class $c$. The probability of classifying

an image can be calculated as

$$p^{\text{s}}(y|\mathbf{x}) = \frac{\exp\left(f_y^{\text{s}}(\mathbf{x})/\tau\right)}{\sum_{c=1}^{C} \exp\left(f_c^{\text{s}}(\mathbf{x})/\tau\right)}, \tag{4}$$

where $f_c^{\text{s}}(\mathbf{x}) = \cos(\mathbf{t}_c^{\text{s}}, \mathbf{I})$ and recall that $\mathbf{I}$ denotes the class feature.

**Perturbation discrete shield (PerturbShield).** As analyzed in Section 3.2, we propose integrating VQ-VAE as a defense module within the prompt tuning framework to mitigate the impact of visual adversarial perturbations. Given the outstanding performance of the Vision Transformer (ViT) (Dosovitskiy et al., 2020) in visual tasks, we use CLIP's ViT as the image encoder. The last Transformer layer of ViT can output two types of image features: the class feature $\mathbf{I} \in \mathbb{R}^{d_v}$ generated by the [class] token and the grid feature $\mathbf{I}_{\text{patch}} \in \mathbb{R}^{N \times d_v}$ from image patches, where $d_v$ denotes the dimension (i.e., 512 for CLIP) and $N$ is the number of patchs.

Since CLIP uses the class feature $\mathbf{I}$ to represent an image, we initially consider reconstructing $\mathbf{I}$ with VQ-VAE. However, this approach has some limitations. That is, the class feature essentially functions as a global patch representation, meaning that we would be restricted to using only a single code from the codebook to represent the entire image. However, the representation capacity of a single code is far inferior to that of multiple codes combined, ultimately resulting in severe information loss. In this case, if an adversarial attack successfully maps the class feature to an incorrect code, it will lead to a wrong prediction.

To address this problem, we propose Perturbation Discrete Shield (PerturbShield), an end-to-end defense module that utilizes discrete representation learning to mitigate adversarial perturbations in VLMs. Specifically, PerturbShield operates through two designed stages: 1) grid-based discrete feature reconstruction; 2) semantic alignment projection, which are introduced in the following.

1) *Grid-based discrete feature reconstruction*. Instead of reconstructing the class feature, we reconstruct the grid feature $\mathbf{I}_{\text{patch}}$ through the VQ-VAE. This process aims to represent the grid feature using a combination of multiple codes to preserve more information. Additionally, the discretization process can help mitigate minor perturbations and enhance the model's robustness as analyzed in Section 3.2. The reconstructed grid feature can be obtained as $\hat{\mathbf{I}}_{\text{patch}} = \text{VQ-VAE}(\mathbf{I}_{\text{patch}})$, where VQ-VAE$(\cdot)$ denotes a VQ-VAE and $\hat{\mathbf{I}}_{\text{patch}} \in \mathbb{R}^{N \times d_v}$.

2) *Semantic alignment projection*. We design a learnable matrix $\mathbf{W}$ that performs two functionalities. First, since $\hat{\mathbf{I}}_{\text{patch}}$ is more robust than $\mathbf{I}$, we could use it to learn a robust representation of the whole image by a linear transformation: $\hat{\mathbf{I}}_{\text{proj}} = \mathbf{W} \cdot \hat{\mathbf{I}}_{\text{patch}}$, where $\mathbf{W} \in \mathbb{R}^{1 \times N}$ and $\hat{\mathbf{I}}_{\text{proj}} \in \mathbb{R}^{d_v}$. Another functionality is to semantically align the transformed latent feature with CLIP's pre-trained embedding space through a feature alignment regularization between $\hat{\mathbf{I}}_{\text{proj}}$ and the class feature $\mathbf{I}$ as

$$\mathcal{L}_{\text{reg}}^{\text{I}} = \|\hat{\mathbf{I}}_{\text{proj}} - \mathbf{I}\|_1, \tag{5}$$

where $\| \cdot \|_1$ denotes the $\ell_1$ norm. These dual functionalities ensure that the adversarial defense process remains aligned with the embedding space of the VLM. Then the prediction probability for an input image $\mathbf{x}$ can be computed as

$$p^{\text{vq}}(y|\mathbf{x}) = \frac{\exp\left(f_y^{\text{vq}}(\mathbf{x})/\tau\right)}{\sum_{c=1}^{C} \exp\left(f_c^{\text{vq}}(\mathbf{x})/\tau\right)}, \tag{6}$$

where $f_c^{\text{vq}}(\mathbf{x}) = \cos(\mathbf{t}_c^{\text{s}}, \hat{\mathbf{I}}_{\text{proj}})$.

**Logits fusion.** There is usually some information loss during the reconstruction process. Consequently, using $\hat{\mathbf{I}}_{\text{proj}}$ alone to compute logits (i.e., the prediction probability) reduces the clean accuracy (e.g., from 19.44% to 17.90% in Figure 3). Since PerturbShield effectively mitigates adversarial attacks, we propose a logits fusion strategy that uses logits from $\hat{\mathbf{I}}_{\text{proj}}$ and $\mathbf{t}^{\text{s}}$ to counter adversarial attacks, while using logits from $\mathbf{I}$ and $\mathbf{t}^{\text{s}}$ to maintain the clean accuracy, thus achieving a better trade-off between the robustness and accuracy. Specifically, the former logits are defined in Eq. (6), and the latter ones are defined in Eq. (4). Then, the fused logits are calculated as

$$p(y|\mathbf{x}) = (p^{\text{vq}}(y|\mathbf{x}) + p^{\text{s}}(y|\mathbf{x}))/2. \tag{7}$$

**Prompt alignment regularization.** To enhance the generalization of learnable prompts and prevent overfitting (Zhou et al., 2022a), we use prompt alignment regularization for learnable prompts. This ensures the consistency between text features generated by learnable prompts and hand-crafted prompts, thereby improving adaptability while preserving CLIP's inherent zero-shot inference capabilities. Specifically, the prompt alignment regularization is formulated as

$$\mathcal{L}_{\text{reg}}^{\text{T}} = \|\mathbf{t}^{\text{s}} - \mathbf{t}\|_1. \tag{8}$$

Moreover, we find that using hand-crafted prompts to regularize learnable prompts as in Eq. (8) can enhance the model's robustness, which is detailed in Section 4.3.

**Training objective.** Based on Eq. (7), the cross-entropy loss is defined as $\mathcal{L}_{\text{ce}}(\mathbf{x}, \mathbf{t}^{\text{s}}, \mathbf{y}) = -\sum_{i=1}^{C} y_i \log p(y_i|\mathbf{x})$. To learn a robust prompt, Li et al. (2024); Zhou et al. (2024) combines prompt tuning with adversarial training. Following this paradigm, we use the learnable prompts updated in the previous epoch to generate adversarial examples dynamically in each epoch as

$$\mathbf{x}_{\text{a}} = \arg\max_{\mathbf{x}_{\text{a}}} \mathcal{L}_{\text{ce}}(\mathbf{x}_{\text{a}}, \mathbf{t}^{\text{s}}, \mathbf{y}), \text{ s.t. } \| \mathbf{x}_{\text{a}} - \mathbf{x} \|_p \leqslant \epsilon. \tag{9}$$

Finally, based on those adversarial examples, the overall training objective of the DEFEAT method is formulated as

$$\mathcal{L}(\mathbf{x}_{\text{a}}, \mathbf{t}^{\text{s}}, \mathbf{t}, \mathbf{y}) = \mathcal{L}_{\text{ce}}(\mathbf{x}_{\text{a}}, \mathbf{t}^{\text{s}}, \mathbf{y}) + \mathcal{L}_{\text{VQ-VAE}}(\mathbf{I}_{\text{patch}}) + \lambda\mathcal{L}_{\text{reg}}^{\text{I}} + \mu\mathcal{L}_{\text{reg}}^{\text{T}}. \tag{10}$$

Due to page limit, the whole algorithm for the proposed DEFEAT method is shown in Appendix A.

## 4 EXPERIMENTS

### 4.1 SETUPS

**Datasets.** By following APT (Li et al., 2024) and CoOp (Zhou et al., 2022b), experiments are conducted under three settings, including adversarial few-shot classification, adversarial cross-dataset generalization, and adversarial domain generalization. The first two settings are performed on 11 image classification datasets, including ImageNet (Deng et al., 2009) and Caltech101 (Fei-Fei et al., 2004) for generic object classification, OxfordPets (Parkhi et al., 2012), StanfordCars (Krause et al., 2013), Flowers102 (Nilsback & Zisserman, 2008), Food101 (Bossard et al., 2014), and FGVCAircraft (Maji et al., 2013) for fine-grained visual categorization, SUN397 (Xiao et al., 2010) for scene recognition, DTD (Cimpoi et al., 2014) for texture classification, EuroSAT (Helber et al., 2019) for satellite image classification, and UCF101 (Soomro et al., 2012) for action recognition. Adversarial domain generalization is conducted on the ImageNet dataset and its variants, including ImageNetV2 (Recht et al., 2019), ImageNet-Sketch (Wang et al., 2019), ImageNet-A (Hendrycks et al., 2021b), and ImageNet-R (Hendrycks et al., 2021a).

**Baselines.** To demonstrate the effectiveness of the proposed DEFEAT method, we compare it against three baseline methods, including zero-shot CLIP (Radford et al., 2021), the textual prompt tuning method (i.e., Adversarial Prompt Tuning (APT) (Li et al., 2024)), and the multi-modal prompt tuning method (i.e., Few-shot Adversarial Prompt learning (FAP) (Zhou et al., 2024)).

**Implementation details.** Our implementation is based on CoOp (Zhou et al., 2022b) and APT (Li et al., 2024). For all baseline methods, we maintain the same experimental settings (e.g., training epochs, training schedules, and data augmentation settings) as specified in their original implementations. All experiments are conducted with a ViT-B/32 CLIP model. Following APT (Li et al., 2024), the image encoder weights for all baselines were pre-trained using TeCoA (Mao et al., 2023). The length of the prompt vectors is fixed to 16. For the DEFEAT methods, $\alpha$, $\beta$, $\lambda$, and $\mu$ are set to 0.5, 0.1, 10, and 20, respectively, across all experiments. For adversarial training and evaluation, we employ the PGD attack (Madry et al., 2018) under the $\ell_\infty$ threat model. Following Li et al. (2024); Mao et al. (2023); Schlarmann et al. (2024), two perturbation sizes, $\epsilon = 1/255$ and $\epsilon = 4/255$, are used. During training, we use 3 steps with a step size of $2\epsilon/3$, and for evaluation, 100 steps with a step size of $\epsilon/4$ and a random start. More details are provided in Appendix B.

### 4.2 MAIN RESULTS

In this section, we report experimental results under different settings.

Table 1: The average performance on the 11 datasets for different $\epsilon$ and shots under the adversarial few-shot classification setting. 'H' denotes the harmonic mean accuracy.

| $\epsilon$ | Method | 1 shot | | | 4 shots | | | 16 shots | | |
|---|---|---|---|---|---|---|---|---|---|---|
| | | Acc. | Rob. | H | Acc. | Rob. | H | Acc. | Rob. | H |
| 1/255 | CLIP | 46.06 | 32.98 | 38.44 | 46.06 | 32.98 | 38.44 | 46.06 | 32.98 | 38.44 |
| | APT | 46.99 | 33.36 | 39.02 | 58.19 | 41.34 | 48.34 | 65.41 | 47.88 | 55.29 |
| | FAP | **52.72** | 37.15 | 43.59 | **59.77** | 42.21 | 49.48 | **66.40** | 48.86 | 56.30 |
| | DEFEAT | 52.07 | **38.99** | **44.59** | 58.37 | **53.04** | **55.58** | 65.03 | **60.77** | **62.83** |
| 4/255 | CLIP | 33.67 | 10.79 | 16.34 | 33.67 | 10.79 | 16.34 | 33.67 | 10.79 | 16.34 |
| | APT | 32.97 | 11.62 | 17.18 | 42.29 | 14.40 | 21.48 | 51.08 | 20.26 | 29.01 |
| | FAP | **38.16** | 13.44 | 19.88 | **44.63** | 15.34 | 22.83 | 50.50 | 19.82 | 28.47 |
| | DEFEAT | 35.78 | **14.73** | **20.87** | 44.13 | **25.28** | **32.15** | 50.98 | **36.05** | **42.23** |

**Adversarial few-shot classification.** In this scenario, we assess the model's capability to develop robust representations from limited labeled data. Specifically, models are tuned using 1, 4, 16 shots per class and evaluated on the remaining samples. The average performance of DEFEAT and baselines on 11 datasets is shown in Table 1, where 'H' denotes the harmonic mean accuracy of robustness and accuracy, measuring the trade-off between these metrics. DEFEAT consistently outperforms zero-shot CLIP, even with 1-shot tuning, achieving significant improvements in accuracy, robustness, and 'H'. For example, under $\epsilon = 1/255$, DEFEAT provides gains of 6.01%, 6.01%, and 6.15% in accuracy, robustness, and 'H', respectively. Furthermore, these improvements scale with the number of shots. With 16-shot tuning, DEFEAT achieves gains of 18.97% (17.31%), 27.79% (25.26%), and 24.39% (25.89%) in accuracy, robustness, and 'H' for $\epsilon = 1/255$ (4/255), respectively.

Existing adversarial prompt tuning methods (i.e., APT and FAP) improve performance over CLIP, confirming the effect of combining adversarial training with prompt tuning. Compared to those methods, DEFEAT performs comparably to the current state-of-the-art method FAP in terms of the accuracy, while consistently outperforms both APT and FAP in terms of the robustness and 'H' across all shot and perturbation sizes. Notably, with 16-shot training under $\epsilon = 4/255$, DEFEAT surpasses APT (FAP) by 15.79% (16.23%) in robustness, and 13.22% (13.76%) in 'H', respectively. Detailed results for each dataset are provided in Appendix C.1.

**Adversarial domain generalization.** We assess the model's capability to generalize to out-of-distribution (OOD) data in the adversarial domain generalization setting. Specifically, models are tuned using 16-shot and 100-shot samples from each of the 1000 classes on ImageNet (source), and then evaluated on four different target domains (i.e., ImageNetV2, ImageNet-Sketch, ImageNet-A, and ImageNet-R). Table 2 shows that all baselines achieve comparable accuracy on ImageNet, as they use the same robust image encoder (Mao et al., 2023) pre-trained on the full ImageNet training set. Nevertheless, with 16-shot training, DEFEAT surpasses CLIP, APT, and FAP in terms of the robustness by 8.92%, 7.04%, and 7.0%, respectively. Across the four target domains, DEFEAT achieves good robustness and 'H', outperforming the strongest baseline (i.e., APT) by 3.31% in robustness and 3.36% in 'H'. Those results confirm that the robustness learned by DEFEAT can be effectively transferred to OOD data. Furthermore, as the number of training samples increases from 16 shots to 100 shots, DEFEAT demonstrates consistent improvements in terms of the robustness, achieving +3.31% on ImageNet and an average of +1.94% across the target domains, which highlights the scalability of our approach. In contrast, the two prompt tuning baselines (i.e., APT and FAP) show little to no improvement.

Table 2: Performance under adversarial domain generalization setting. $\epsilon = 4/255$.

| | Method | Source | | | Target | | | | | | | | | | | | | |
|---|---|---|---|---|---|---|---|---|---|---|---|---|---|---|---|---|---|---|
| | | ImageNet | | | ImageNet-V2 | | | ImageNet-Sketch | | | ImageNet-A | | | ImageNet-R | | | **Average** | | |
| | | Acc. | Rob. | H | Acc. | Rob. | H | Acc. | Rob. | H | Acc. | Rob. | H | Acc. | Rob. | H | Acc. | Rob. | H |
| zero shot | CLIP | 40.11 | 10.14 | 16.19 | 33.11 | 7.49 | 12.22 | 17.59 | 7.24 | 10.26 | 4.04 | 0.28 | 0.52 | 37.52 | 12.51 | 18.76 | 23.07 | 6.88 | 10.60 |
| 16 shots | APT | **41.06** | 12.02 | 18.60 | **33.67** | 9.09 | 14.32 | 18.22 | 7.87 | 10.99 | **4.19** | 0.36 | 0.66 | 37.04 | 13.29 | 19.56 | **23.28** | 7.65 | 11.52 |
| | FAP | 40.32 | 12.06 | 18.57 | 32.81 | 9.17 | 14.33 | 16.42 | 7.11 | 9.92 | 3.87 | 0.43 | 0.77 | 36.04 | 13.55 | 19.70 | 22.29 | 7.57 | 11.30 |
| | DEFEAT | 40.68 | **19.06** | **25.96** | 33.12 | **14.96** | **20.61** | 18.27 | **9.87** | **12.82** | 3.97 | **0.88** | **1.44** | 37.35 | **18.14** | **24.42** | 23.18 | **10.96** | **14.88** |
| 100 shots | APT | 40.35 | 12.11 | 18.63 | 33.38 | 9.19 | 14.41 | **18.86** | 8.16 | 11.39 | **4.21** | 0.40 | 0.73 | **37.81** | 13.70 | 20.11 | **23.57** | 7.86 | 11.79 |
| | FAP | **40.76** | 12.35 | 18.96 | **33.60** | 9.14 | 14.37 | 16.41 | 6.98 | 9.79 | 3.65 | 0.39 | 0.70 | 36.36 | 13.60 | 19.80 | 22.51 | 7.53 | 11.28 |
| | DEFEAT | 40.23 | **22.71** | **29.03** | 32.77 | **16.44** | **21.90** | 18.49 | **11.53** | **14.20** | 4.20 | **1.37** | **2.07** | 37.34 | **21.45** | **27.25** | 23.20 | **12.70** | **16.41** |

Table 3: Performance under adversarial cross-dataset generalization setting. $\epsilon = 4/255$.

| 16 shots | Method | Source | Target | | | | | | | | | | |
|---|---|---|---|---|---|---|---|---|---|---|---|---|---|
| | | ImageNet | Caltech101 | OxfordPets | StanfordCars | Flowers102 | Food101 | FGVCAircraft | SUN397 | DTD | EuroSAT | UCF101 | Avg. |
| Acc. | CLIP | 40.11 | 78.78 | 66.26 | 10.32 | 30.13 | 23.52 | 7.17 | 33.24 | 24.29 | 19.56 | 36.98 | 33.03 |
| | APT | 41.06 | 79.63 | 64.92 | 11.55 | 27.24 | 23.58 | 4.77 | 30.87 | 22.22 | 16.31 | 33.68 | 31.48 |
| | FAP | 40.32 | 79.19 | 63.67 | 9.64 | 30.25 | 22.82 | 5.43 | 31.92 | 23.23 | 16.79 | 32.25 | 31.52 |
| | DEFEAT | 40.68 | 78.22 | 66.69 | 9.40 | 31.10 | 22.01 | 6.09 | 32.28 | 23.94 | 17.95 | 33.15 | 32.08 |
| Rob. | CLIP | 10.14 | 43.61 | 15.56 | 0.99 | 8.93 | 3.27 | 0.36 | 6.20 | 11.35 | 11.22 | 7.01 | 10.85 |
| | APT | 12.02 | 46.86 | 21.10 | 1.49 | 8.89 | 3.75 | 0.60 | 6.85 | 9.93 | 11.15 | 7.56 | 11.82 |
| | FAP | 12.06 | 46.25 | 20.71 | 1.55 | 9.74 | 3.67 | 0.66 | 7.15 | 10.82 | 11.38 | 7.45 | 11.94 |
| | DEFEAT | 19.06 | 54.32 | 31.97 | 3.84 | 18.80 | 8.72 | 2.34 | 13.47 | 15.60 | 12.89 | 13.59 | 17.55 |
| H | CLIP | 16.19 | 56.14 | 25.20 | 1.81 | 13.78 | 5.74 | 0.69 | 10.45 | 15.47 | 14.26 | 11.79 | 16.33 |
| | APT | 18.60 | 59.00 | 31.85 | 2.64 | 13.41 | 6.47 | 1.07 | 11.21 | 13.73 | 13.25 | 12.35 | 17.18 |
| | FAP | 18.57 | 58.40 | 31.25 | 2.67 | 14.74 | 6.32 | 1.18 | 11.68 | 14.76 | 13.57 | 12.10 | 17.32 |
| | DEFEAT | 25.96 | 64.12 | 43.22 | 5.45 | 23.43 | 12.49 | 3.38 | 19.01 | 18.89 | 15.00 | 19.28 | 22.69 |
| 100 shots | Method | ImageNet | Caltech101 | OxfordPets | StanfordCars | Flowers102 | Food101 | FGVCAircraft | SUN397 | DTD | EuroSAT | UCF101 | Avg. |
| Acc. | CLIP | 40.11 | 78.78 | 66.26 | 10.32 | 30.13 | 23.52 | 7.17 | 33.24 | 24.29 | 19.56 | 36.98 | 33.03 |
| | APT | 40.35 | 80.28 | 64.60 | 11.76 | 29.03 | 23.87 | 5.85 | 32.91 | 25.71 | 14.17 | 34.87 | 32.31 |
| | FAP | 40.76 | 75.54 | 64.30 | 10.21 | 29.76 | 23.37 | 3.93 | 30.49 | 24.94 | 22.01 | 31.51 | 31.61 |
| | DEFEAT | 40.23 | 76.63 | 65.74 | 9.46 | 30.25 | 20.02 | 6.51 | 31.31 | 23.94 | 16.67 | 31.69 | 31.22 |
| Rob. | CLIP | 10.14 | 43.61 | 15.56 | 0.99 | 8.93 | 3.27 | 0.36 | 6.20 | 11.35 | 11.22 | 7.01 | 10.85 |
| | APT | 12.11 | 46.82 | 21.18 | 1.52 | 9.66 | 3.76 | 0.69 | 7.20 | 12.41 | 11.06 | 8.33 | 12.26 |
| | FAP | 12.35 | 44.50 | 20.47 | 1.47 | 9.50 | 3.89 | 0.39 | 6.91 | 10.17 | 12.19 | 7.35 | 11.68 |
| | DEFEAT | 22.71 | 59.35 | 42.95 | 5.58 | 21.52 | 11.91 | 3.78 | 16.29 | 16.49 | 12.98 | 16.44 | 20.73 |
| H | CLIP | 16.19 | 56.14 | 25.20 | 1.81 | 13.78 | 5.74 | 0.69 | 10.45 | 15.47 | 14.26 | 11.79 | 16.33 |
| | APT | 18.63 | 59.15 | 31.90 | 2.69 | 14.50 | 6.50 | 1.23 | 11.82 | 16.74 | 12.42 | 13.45 | 17.78 |
| | FAP | 18.96 | 56.01 | 31.05 | 2.57 | 14.40 | 6.67 | 0.71 | 11.27 | 14.45 | 15.69 | 11.92 | 17.06 |
| | DEFEAT | 29.03 | 66.89 | 51.96 | 7.02 | 25.15 | 14.94 | 4.78 | 21.43 | 19.53 | 14.60 | 21.65 | 24.92 |

**Adversarial cross-dataset generalization.** We assess the model's zero-shot adversarial robustness in the adversarial cross-dataset generalization setting. Models are tuned using 16-shot and 100-shot samples from each of the 1000 classes on ImageNet and evaluated on the other 10 datasets. Table 3 shows that DEFEAT achieves competitive accuracy and the highest robustness and 'H' compared to all baseline methods across all datasets, including both source and target domains. Specifically, with 16-shot tuning, DEFEAT outperforms the strongest baseline (i.e., FAP) by an average of 5.61% in terms of robustness and 5.37% in terms of 'H'. Similar to the adversarial domain generalization setting, APT and FAP show no significant performance improvements when the number of training samples increases from 16 shots to 100 shots. In contrast, DEFEAT demonstrates scalability, achieving gains of 3.65% in terms of robustness on ImageNet and 3.18% in terms of average robustness on 10 target datasets. Those results demonstrate the good zero-shot adversarial robustness of DEFEAT.

## 4.3 Ablation Studies

In this section, we conduct ablation studies to analyze the effectiveness of the PerturbShield module, logits fusion strategy, and hyperparameter (i.e., $\mu$). Experiments are conducted using 16 examples per class on the EuroSAT dataset under the adversarial few-shot classification setting.

**Effects of components.** Table 4 shows the ablation study for PerturbShield and logits fusion strategy. Due to information loss during the reconstruction process, using PerturbShield alone without logits fusion results in limited performance improvements on CLIP and can negatively impact DEFEAT. Unlike CLIP, which computes logits using hand-crafted prompts, DEFEAT relies on learnable prompts. The information loss can impair prompt learning, thereby causing adverse effects.

By combining PerturbShield with the logits fusion strategy, adversarial attacks can be mitigated effectively while maintaining good clean accuracy, thus achieving a better trade-off between accuracy and robustness. Specifically, CLIP combining Perturb-Shield and logits fusion strategy outperforms the original CLIP by 9.91% in 'H' (24.13% v.s. 14.22%), and DEFEAT with this combination outperforms its counterpart without it by 20.38% in 'H' (56.69% v.s. 36.31%). Therefore, we combine PerturbShield with the logits fusion strategy.

Table 4: Ablation study. $\epsilon = 4/255$.

| Baselines | Adversarial prompt tuning | PerturbShield | Logits fusion | Acc. | Rob. | H |
|---|---|---|---|---|---|---|
| CLIP | ✗ | ✗ | ✗ | 19.44 | 11.21 | 14.22 |
| | ✗ | ✓ | ✗ | 17.90 | 12.90 | 14.99 (+0.77) |
| | ✗ | ✓ | ✓ | 25.53 | 22.88 | 24.13 (+9.91) |
| DEFEAT | ✓ | ✗ | ✗ | 64.12 | 25.33 | 36.31 |
| | ✓ | ✓ | ✗ | 22.21 | 17.48 | 19.56 (-16.75) |
| | ✓ | ✓ | ✓ | 58.88 | 54.65 | 56.69 (+20.38) |

**Effect of $\mu$.** $\mu$ controls the weight of $\mathcal{L}_{\text{reg}}^{\text{T}}$, which apply regularization from learnable prompts to hand-crafted prompts. According to Figure 5, we can see that applying regularization (i.e., $\mu > 0$) of learnable prompts can enhance the model's robustness. As $\mu$ increases, the robustness of DEFEAT improves, achieving the best overall performance at $\mu = 20$. Therefore, we set $\mu = 20$ in our experiments.

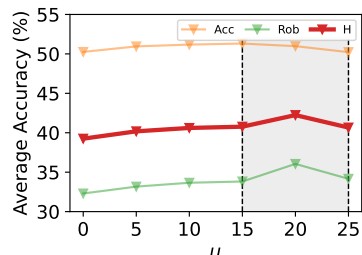

Figure 5: The average performance of DEFEAT on all datasets w.r.t. different $\mu$ values.

**How does DEFEAT enhance adversarial robustness?** To answer this question, we visualize the image feature (i.e., class feature) output by the image encoder using t-SNE. Figure 6 shows that CLIP exhibits a significant distribution shift between clean and adversarial examples, while adversarial prompt tuning methods (i.e., APT and FAP) reduce this shift. Notably, the image encoder remains frozen during training and testing for all methods, demonstrating that adversarial prompt tuning can mitigate the backward adversarial gradient, thereby enhancing model robustness. DEFEAT not only incorporates adversarial prompt tuning but also further mitigates the backward adversarial gradient through PerturbShield. So the distribution shift of latent features between clean and adversarial examples is further reduced compared to APT and FAP, providing additional evidence of the effectiveness of the proposed DEFEAT method.

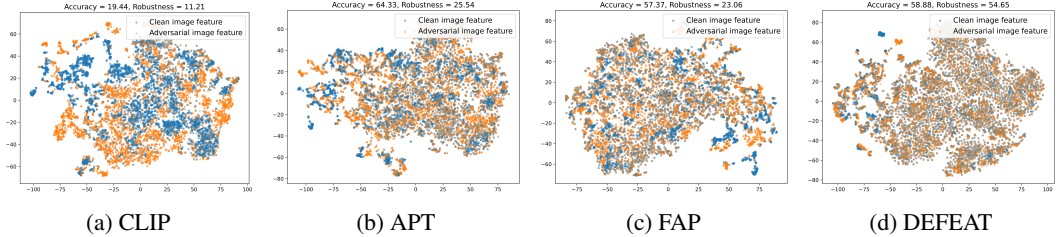

| (a) CLIP | (b) APT | (c) FAP | (d) DEFEAT |

Figure 6: Visualization of clean and adversarial examples on the EuroSAT dataset. Models are trained with 16-shot samples per class. $\epsilon = 4/255$.

*Moreover, due to page limit, more experiments can be found in Appendix C, including evaluations against stronger attacks and a custom adaptive attack designed for DEFEAT, generalization to alternative backbones, detailed hyperparameter analysis, and computational cost analysis.*

## 5 CONCLUSION

This work presents a robust prompt tuning framework for VLMs. We begin by analyzing the mitigating effect of feature discretization on visual adversarial perturbation. Building on this, we propose the **D**iscrete Lat**E**nt **F**eatur**E** based **A**dversarial **T**raining (**DEFEAT**) method, which ablates adversarial attacks in the feature representation. Specifically, DEFEAT introduces a perturbation discrete shield module that reconstructs discrete latent features, and designs a logits fusion strategy to improve the trade-off between robustness and accuracy. Moreover, DEFEAT integrates prompt tuning with adversarial training and applies prompt alignment regularization, further enhancing the adversarial robustness. Extensive experiments on 15 datasets demonstrate that DEFEAT achieves state-of-the-art performance.

## ACKNOWLEDGMENTS

This work was supported by National Natural Science Foundation of China under Grant no. 62136005 and Shenzhen fundamental research program JCYJ20250604144724032.

ETHICS STATEMENT

This paper presents work whose goal is to advance the field of Machine Learning by improving the safety and reliability of AI systems. The authors have read and comply with the ICLR Code of Ethics. The research did not involve human subjects, animal experiments, or personally identifiable data. All experiments were conducted on publicly available benchmarks and open-source models. We have carefully considered the broader impacts and believe that this work poses no foreseeable risks of harm while contributing to the development of robust and secure vision-language models.

REPRODUCIBILITY STATEMENT

The authors have made significant efforts to ensure the reproducibility of results. Section 4.1 details the experimental setup, including datasets, model configurations, and hyperparameter settings. Additional ablations in Section 4.3 and Appendix C.2 further analyze the effect of each module and hyperparameters. During the reviewing process, the source code is supplied anonymously as part of the supplementary materials. Additionally, upon the acceptance of the paper, this code will be publicly released.

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

# Contents of the Appendix

## A  ALGORITHM

The whole algorithm for the proposed DEFEAT methods is shown in Algorithm 1.

---

**Algorithm 1** Discrete Latent Feature based Adversarial Training (DEFEAT)

---

1: **Input:** Dataset $\mathbb{D}$, learnable prompt vectors $\mathbf{v}$, textual embeddings of hand-crafted prompts $\mathbf{t}$, CLIP pre-trained image encoder $\mathcal{I}$ and text encoder $\mathcal{T}$, VQ-VAE model $\boldsymbol{\theta}_{\text{VQ-VAE}}$, learnable matrix $\mathbf{W}$, weight parameters $\lambda$, $\mu$, learning rate $\eta$, and perturbation size $\epsilon$.

2: **for all** training epochs **do**

3:     **for all** $\mathbf{x}$, $\mathbf{y} \in$ minibatch **do**

4:         Use $\mathbf{v}$ and [CLS] to generate textual embeddings of learnable prompts $\mathbf{t}^{\text{s}}$;

5:         Generate adversarial examples
$$\mathbf{x}_{\text{a}} \leftarrow \arg\max_{\mathbf{x}_{\text{a}}} \mathcal{L}_{\text{ce}}(\mathbf{x}_{\text{a}}, \mathbf{t}^{\text{s}}, \mathbf{y}), \ \text{s.t.} \parallel \mathbf{x}_{\text{a}} - \mathbf{x} \parallel_p \leqslant \epsilon;$$

6:         Feed $\mathbf{x}_{\text{a}}$ to $\mathcal{I}$ to generate class feature $\mathbf{I}$ and grid feature $\mathbf{I}_{\text{patch}}$;

7:         Reconstruct $\mathbf{I}_{\text{patch}}$ using a VQ-VAE, obtaining $\hat{\mathbf{I}}_{\text{patch}}$;

8:         Obtain a robust representation by a transformation
$$\hat{\mathbf{I}}_{\text{proj}} \leftarrow \mathbf{W} \cdot \hat{\mathbf{I}}_{\text{patch}}$$

9:         Fuse the logits from $\hat{\mathbf{I}}_{\text{proj}}$ in Eq. (6) and $\mathbf{I}$ in Eq. (4)
$$p(y|\mathbf{x}_{\text{a}}) \leftarrow (p^{\text{vq}}(y|\mathbf{x}_{\text{a}}) + p^{\text{s}}(y|\mathbf{x}_{\text{a}}))/2;$$

10:        Train with the objective
$$\mathcal{L}(\mathbf{x}_{\text{a}}, \mathbf{t}^{\text{s}}, \mathbf{t}, \mathbf{y}) \leftarrow \mathcal{L}_{\text{ce}}(\mathbf{x}_{\text{a}}, \mathbf{t}^{\text{s}}, \mathbf{y}) + \mathcal{L}_{\text{VQ-VAE}}(\mathbf{I}_{\text{patch}}) + \lambda\mathcal{L}_{\text{reg}}^{\text{I}} + \mu\mathcal{L}_{\text{reg}}^{\text{T}},$$
        where $\mathcal{L}_{\text{ce}}(\mathbf{x}_{\text{a}}, \mathbf{t}^{\text{s}}, \mathbf{y}) \leftarrow -\sum_{i=1}^{C} y_i \log p(y_i|\mathbf{x}_{\text{a}})$, $\mathcal{L}_{\text{reg}}^{\text{I}}$ is defined in Eq. (5), $\mathcal{L}_{\text{reg}}^{\text{T}}$ is defined in Eq. (8), and $\mathcal{L}_{\text{VQ-VAE}}(\cdot)$ is defined in Eq. (3);

11:        Upadate the learnable parameters
$$\mathbf{v} \leftarrow \mathbf{v} - \eta\nabla_{\mathbf{v}}\mathcal{L}(\mathbf{x}_{\text{a}}, \mathbf{t}^{\text{s}}, \mathbf{t}, \mathbf{y});$$
$$\mathbf{W} \leftarrow \mathbf{W} - \eta\nabla_{\mathbf{W}}\mathcal{L}(\mathbf{x}_{\text{a}}, \mathbf{t}^{\text{s}}, \mathbf{t}, \mathbf{y});$$
$$\boldsymbol{\theta}_{\text{VQ-VAE}} \leftarrow \boldsymbol{\theta}_{\text{VQ-VAE}} - \eta\nabla_{\boldsymbol{\theta}_{\text{VQ-VAE}}}\mathcal{L}(\mathbf{x}_{\text{a}}, \mathbf{t}^{\text{s}}, \mathbf{t}, \mathbf{y}).$$

12:     **end for**

13: **end for**

---

## B  IMPLEMENTATION DETAILS

All experiments are conducted on NVIDIA GeForce RTX 3090, except for the ImageNet dataset, which is on Quadro RTX 8000. Training is conducted using SGD with an initial learning rate of 0.002, which is decayed using the cosine annealing rule. The maximum epoch is set to 200, 100, and 50 for 16, 4, and 1 shots, respectively. For ImageNet, they are 50, 20, and 20. A warm-up strategy is used by fixing the learning rate to $10^{-5}$ during the first epoch. The VQ-VAE within the PerturbShield module is configured with an input dimension of 512 to align with CLIP ViT-B/32 grid features, projecting them into a latent dimension of 256 using a codebook of 512 unique embedding vectors. The encoder of VQ-VAE consists of two convolutional layers followed by a residual block, while the decoder of VQ-VAE mirrors this structure with a residual block and a transposed convolutional layer for feature reconstruction.

The hand-crafted prompts for different datasets used in the prompt alignment regularization follow Radford et al. (2021); Zhou et al. (2022b) and are shown below:

```
ImageNet: "a photo of a [CLS]."
Caltech101: "a photo of a [CLS]."
OxfordPets: "a photo of a [CLS], a type of pet."
StanfordCars: "a photo of a [CLS]."
OxfordFlowers: "a photo of a [CLS], a type of flower."
Food101: "a photo of [CLS], a type of food."
FGVCAircraft: "a photo of a [CLS], a type of aircraft."
SUN397: "a photo of a [CLS]."
DTD: "a photo of a [CLS], a type of texture."
EuroSAT: "a centered satellite photo of [CLS]."
UCF101: "a photo of a person doing [CLS]."
```

Note that [CLS] denotes the placeholder for the class name.

## C  ADDITIONAL EXPERIMENTAL RESULTS

### C.1  COMPREHENSIVE RESULTS OF ADVERSARIAL FEW-SHOT CLASSIFICATION

In this section, we provide the full results of adversarial few-shot classification performance on 11 datasets. The results in Tables A5 and A6 show that the proposed DEFEAT method can consistently outperform the baseline methods in terms of the robustness and harmonic mean accuracy across nearly all datasets and shot settings. For each specific dataset, DEFEAT consistently outperforms CLIP across all shots. For adversarial prompt tuning methods, DEFEAT outperforms APT and FAP across almost all the datasets and shot settings, with particularly notable improvements on EuroSAT, UCF101, etc.. For example, with 16-shot training, DEFEAT surpasses APT and FAP by 20.13% and 23.79% in terms of 'H' (i.e., 56.69% vs. 36.56% vs. 32.90%) on EuroSAT, and by 23.21% and 21.35% (i.e., 48.45% vs. 25.24% vs. 27.10%) on UCF101.

Figure A7 shows that as the number of shots increases, DEFEAT achieves greater improvements compared to APT and FAP. Those results demonstrate DEFEAT's effectiveness and scalability.

### C.2  ANALYSIS ON MORE HYPERPARAMETER SENSITIVITY

In this section, we conduct ablation studies to analyse the effectiveness of hyperparameters (i.e., $\alpha$, $\beta$, and $\lambda$).

**Effect of $\alpha$.** In Eq. (3), $\alpha$ controls the weight of the reconstruction loss, which is used to optimize the decoder and the encoder of VQ-VAE. According to Figure A8a, we can see that DEFEAT is insensitive within the range [0,5]. We set $\alpha = 0.5$ in our experiments.

**Effect of $\beta$.** In Eq. (3), $\beta$ controls the weight of the codebook loss, which is used to learn the embedding space of the codebook. According to Figure A8b, applying the codebook loss (i.e., $\beta > 0$) results in better overall performance compared to excluding it (i.e., $\beta > 0$). Failure to update the codebook reduces the representational capacity of the embeddings, thereby increasing the reconstruction quantization error of the VQ-VAE. Additionally, the model may collapse if the codebook is poorly initialized. The overall performance of DEFEAT is insensitive within the range $\beta \in [0.01, 0.1]$. In our experiments, we set $\beta = 0.1$.

Table A5: The performance on the 11 datasets for different shots under the adversarial few-shot classification setting when $\epsilon = 1/255$, where 'H' denotes the harmonic mean accuracy.

| Dataset | $\epsilon = 1/255$ | CLIP | 1 shot | | | 4 shot | | | 16 shot | | |
|---|---|---|---|---|---|---|---|---|---|---|---|
| | | | APT | FAP | DEFEAT | APT | FAP | DEFEAT | APT | FAP | DEFEAT |
| **Average** | Acc. | 46.06 | 46.99 | **52.72** | 52.07 | 58.19 | **59.77** | 58.37 | 65.41 | **66.40** | 65.03 |
| | Rob. | 32.98 | 33.36 | 37.15 | **38.99** | 41.34 | 42.21 | **53.04** | 47.88 | 48.86 | **60.77** |
| | H | 38.44 | 39.02 | 43.58 | **44.59** | 48.34 | 49.48 | **55.57** | 55.29 | 56.29 | **62.83** |
| ImageNet | Acc. | 55.34 | **55.48** | 55.01 | 54.33 | 56.31 | **57.39** | 56.11 | 50.73 | **58.46** | 57.54 |
| | Rob. | 38.48 | 39.00 | 37.79 | **39.68** | 39.30 | 39.63 | **41.36** | 33.37 | 40.37 | **53.07** |
| | H | 45.40 | 45.80 | 44.80 | **45.86** | 46.29 | 46.88 | **47.62** | 40.26 | 47.76 | **55.21** |
| Caltech101 | Acc. | 85.92 | 86.46 | 88.15 | **89.66** | 89.17 | 90.87 | **91.16** | 92.90 | 92.21 | **93.39** |
| | Rob. | 75.21 | 77.32 | 78.86 | **80.73** | 79.68 | 80.81 | **87.99** | 83.98 | 84.30 | **91.12** |
| | H | 80.21 | 81.63 | 83.25 | **84.96** | 84.16 | 85.55 | **89.55** | 88.22 | 88.08 | **92.24** |
| OxfordPets | Acc. | 79.48 | 78.33 | **79.56** | 77.19 | 83.48 | 83.21 | **84.08** | 83.95 | **85.55** | 84.71 |
| | Rob. | 63.26 | 61.57 | 60.56 | **62.36** | 66.18 | 66.34 | **73.37** | 65.14 | 69.15 | **79.53** |
| | H | 70.45 | 68.95 | 68.77 | **68.99** | 73.83 | 73.82 | **78.36** | 73.36 | 76.48 | **82.04** |
| StanfordCars | Acc. | 25.21 | 41.09 | **45.55** | 42.87 | 49.51 | **51.14** | 47.69 | **60.20** | 57.54 | 56.42 |
| | Rob. | 12.50 | 22.44 | 24.05 | **26.81** | 26.94 | 27.47 | **44.05** | 37.21 | 32.79 | **51.65** |
| | H | 16.71 | 29.03 | 31.48 | **32.99** | 34.89 | 35.74 | **45.80** | 45.99 | 41.77 | **53.93** |
| Flowers102 | Acc. | 48.15 | 23.63 | **59.60** | 53.39 | **76.61** | 70.36 | 72.72 | **86.80** | 82.46 | 84.90 |
| | Rob. | 32.68 | 17.46 | **42.14** | 40.80 | 59.68 | 51.48 | **69.02** | 73.57 | 66.63 | **81.81** |
| | H | 38.93 | 20.08 | **49.37** | 46.25 | 67.09 | 59.46 | **70.82** | 79.64 | 73.70 | **83.33** |
| Food101 | Acc. | 46.58 | 41.18 | **55.36** | 49.83 | 45.37 | **59.24** | 51.29 | 54.68 | **64.48** | 56.31 |
| | Rob. | 27.38 | 22.12 | **33.32** | 32.07 | 24.95 | 36.20 | **46.84** | 33.02 | 41.19 | **49.45** |
| | H | 34.49 | 28.78 | **41.60** | 39.02 | 32.20 | 44.94 | **48.96** | 41.18 | 50.27 | **52.66** |
| FGVCAircraft | Acc. | 12.51 | 2.01 | **18.69** | 17.55 | 14.64 | 21.00 | **21.27** | **28.74** | 26.55 | 25.74 |
| | Rob. | 6.09 | 1.08 | 10.41 | **11.16** | 7.44 | 11.43 | **19.98** | 16.53 | 14.97 | **23.46** |
| | H | 8.19 | 1.41 | 13.37 | **13.64** | 9.87 | 14.80 | **20.60** | 20.99 | 19.15 | **24.55** |
| SUN397 | Acc. | 48.67 | 51.17 | **53.50** | 52.63 | 56.35 | **58.90** | 58.75 | 62.56 | 62.69 | **62.83** |
| | Rob. | 31.70 | 32.44 | 34.90 | **35.73** | 36.48 | 38.71 | **54.01** | 42.31 | 42.67 | **57.53** |
| | H | 38.39 | 39.71 | 42.24 | **42.56** | 44.29 | 46.72 | **56.28** | 50.48 | 50.78 | **60.06** |
| DTD | Acc. | 31.97 | 34.10 | 36.83 | 38.48 | 47.04 | **48.58** | 47.93 | 55.26 | 56.68 | **57.15** |
| | Rob. | 23.88 | 23.40 | 24.23 | **28.55** | 33.33 | 35.87 | **44.33** | 39.54 | 41.67 | **52.66** |
| | H | 27.34 | 27.75 | 29.23 | **32.78** | 39.02 | 41.27 | **46.06** | 46.10 | 48.03 | **54.81** |
| EuroSAT | Acc. | 23.62 | **51.44** | 31.80 | 42.00 | **61.43** | 55.38 | 47.98 | 74.15 | **74.89** | 67.20 |
| | Rob. | 16.48 | **35.37** | 21.89 | 31.80 | 39.10 | 31.88 | **43.86** | 51.53 | 52.48 | **62.80** |
| | H | 19.41 | **41.92** | 25.93 | 36.20 | **47.78** | 40.47 | 45.83 | 60.80 | 61.71 | **64.93** |
| UCF101 | Acc. | 49.22 | 52.02 | **55.86** | 54.80 | 60.22 | 61.43 | **63.05** | **69.55** | 68.89 | 69.10 |
| | Rob. | 35.13 | 34.73 | **40.47** | 39.15 | 41.63 | 44.49 | **58.58** | 50.52 | 51.23 | **65.40** |
| | H | 41.00 | 41.65 | **46.94** | 45.67 | 49.23 | 51.61 | **60.73** | 58.53 | 58.76 | **67.20** |

Table A6: The performance on the 11 datasets for different shots under the adversarial few-shot classification setting when $\epsilon = 4/255$, where 'H' denotes the harmonic mean accuracy.

| Dataset | $\epsilon = 4/255$ | CLIP | 1 shot | | | 4 shot | | | 16 shot | | |
|---|---|---|---|---|---|---|---|---|---|---|---|
| | | | APT | FAP | DEFEAT | APT | FAP | DEFEAT | APT | FAP | DEFEAT |
| **Average** | Acc. | 33.67 | 32.97 | **38.16** | 35.78 | 42.29 | **44.63** | 44.13 | **51.08** | 50.50 | 50.98 |
| | Rob. | 10.79 | 11.62 | 13.44 | **14.73** | 14.40 | 15.34 | **25.28** | 20.26 | 19.82 | **36.05** |
| | H | 16.34 | 17.19 | 19.88 | **20.87** | 21.49 | 22.83 | **32.14** | 29.02 | 28.47 | **42.24** |
| ImageNet | Acc. | 40.11 | **37.88** | 36.95 | 37.32 | 39.13 | 38.94 | **39.28** | 41.06 | 40.45 | **41.09** |
| | Rob. | 10.14 | 10.79 | 11.41 | **12.48** | 11.49 | 11.86 | **12.66** | 12.02 | 12.03 | **18.42** |
| | H | 16.19 | 16.80 | 17.44 | **18.70** | 17.76 | 18.18 | **19.15** | 18.60 | 18.54 | **25.44** |
| Caltech101 | Acc. | 78.78 | 77.89 | 76.67 | **79.55** | 81.62 | 77.61 | **83.16** | 86.29 | 80.04 | **87.63** |
| | Rob. | 43.61 | 45.84 | 48.36 | **51.03** | 47.59 | 51.81 | **58.74** | 56.75 | 55.42 | **78.30** |
| | H | 56.14 | 57.71 | 59.31 | **62.18** | 60.12 | 62.14 | **68.85** | 68.47 | 65.49 | **82.70** |
| OxfordPets | Acc. | 66.26 | 59.58 | **63.01** | 57.51 | 65.99 | 66.86 | **69.45** | 67.29 | 69.09 | **72.17** |
| | Rob. | 15.56 | 15.54 | 20.31 | **22.16** | 17.80 | 22.59 | **27.42** | 19.98 | 26.17 | **30.77** |
| | H | 25.20 | 24.65 | 30.72 | **31.99** | 28.04 | 33.77 | **39.32** | 30.81 | 37.96 | **43.14** |
| StanfordCars | Acc. | 10.32 | 19.49 | **27.41** | 22.96 | 23.93 | **33.96** | 25.90 | 31.60 | **41.06** | 32.45 |
| | Rob. | 0.99 | 2.90 | 3.27 | **4.74** | 4.44 | 4.76 | **8.20** | 7.70 | 6.33 | **19.51** |
| | H | 1.81 | 5.05 | 5.84 | **7.86** | 7.49 | 8.35 | **12.46** | 12.38 | 10.97 | **24.37** |
| Flowers102 | Acc. | 30.13 | 35.93 | **42.55** | 34.59 | **60.25** | 55.58 | 56.11 | **76.41** | 66.10 | 71.13 |
| | Rob. | 8.93 | 11.49 | **14.21** | 14.01 | 22.45 | 20.46 | **48.60** | 37.52 | 30.82 | **58.26** |
| | H | 13.78 | 17.41 | **21.30** | 19.94 | 32.71 | 29.91 | **52.09** | 50.33 | 42.04 | **64.05** |
| Food101 | Acc. | 23.52 | 20.73 | **33.46** | 26.96 | 21.97 | **38.84** | 29.75 | 30.39 | **45.60** | 33.57 |
| | Rob. | 3.27 | 3.42 | 5.46 | **6.01** | 4.01 | 6.39 | **8.29** | 7.90 | 8.23 | **20.50** |
| | H | 5.74 | 5.87 | 9.39 | **9.83** | 6.78 | 10.97 | **12.97** | 12.54 | 13.94 | **25.46** |
| FGVCAircraft | Acc. | 7.17 | 1.41 | **11.94** | 11.67 | 2.31 | **15.27** | 14.58 | **20.31** | 18.57 | 18.03 |
| | Rob. | 0.36 | 0.42 | 2.52 | **2.73** | 0.51 | 2.73 | **10.26** | 6.15 | 5.07 | **11.94** |
| | H | 0.69 | 0.65 | 4.16 | **4.42** | 0.84 | 4.63 | **12.04** | 9.44 | 7.97 | **14.37** |
| SUN397 | Acc. | 33.24 | 32.32 | **36.44** | 35.68 | 39.06 | 39.68 | **41.29** | 45.21 | 43.66 | **46.26** |
| | Rob. | 6.20 | 5.76 | 8.63 | **9.14** | 7.92 | 9.69 | **19.99** | 11.27 | 11.44 | **28.88** |
| | H | 10.45 | 9.78 | 13.96 | **14.55** | 13.17 | 15.58 | **26.94** | 18.04 | 18.13 | **35.56** |
| DTD | Acc. | 24.29 | 23.29 | **27.78** | 25.24 | 36.17 | 37.41 | **38.18** | **45.86** | 42.85 | 44.15 |
| | Rob. | 11.35 | 9.46 | 10.87 | **11.88** | 14.13 | 15.60 | **26.48** | 21.51 | 20.98 | **32.33** |
| | H | 15.47 | 13.45 | 15.63 | **16.16** | 20.32 | 22.02 | **31.27** | 29.28 | 28.17 | **37.33** |
| EuroSAT | Acc. | 19.56 | 21.09 | **28.02** | 25.49 | **49.98** | 40.58 | 39.47 | **64.33** | 57.37 | 58.88 |
| | Rob. | 11.22 | 14.53 | 13.94 | **15.77** | 16.44 | 8.57 | **28.07** | 25.54 | 23.06 | **54.65** |
| | H | 14.26 | 17.21 | 18.62 | **19.49** | 24.74 | 14.15 | **32.81** | 36.56 | 32.90 | **56.69** |
| UCF101 | Acc. | 36.98 | 33.10 | 35.58 | **36.56** | 44.75 | 46.21 | **48.27** | 53.16 | 50.67 | **55.43** |
| | Rob. | 7.01 | 7.72 | 8.86 | **12.11** | 11.66 | 14.27 | **29.32** | 16.55 | 18.50 | **43.03** |
| | H | 11.79 | 12.52 | 14.19 | **18.19** | 18.50 | 21.81 | **36.48** | 25.24 | 27.10 | **48.45** |

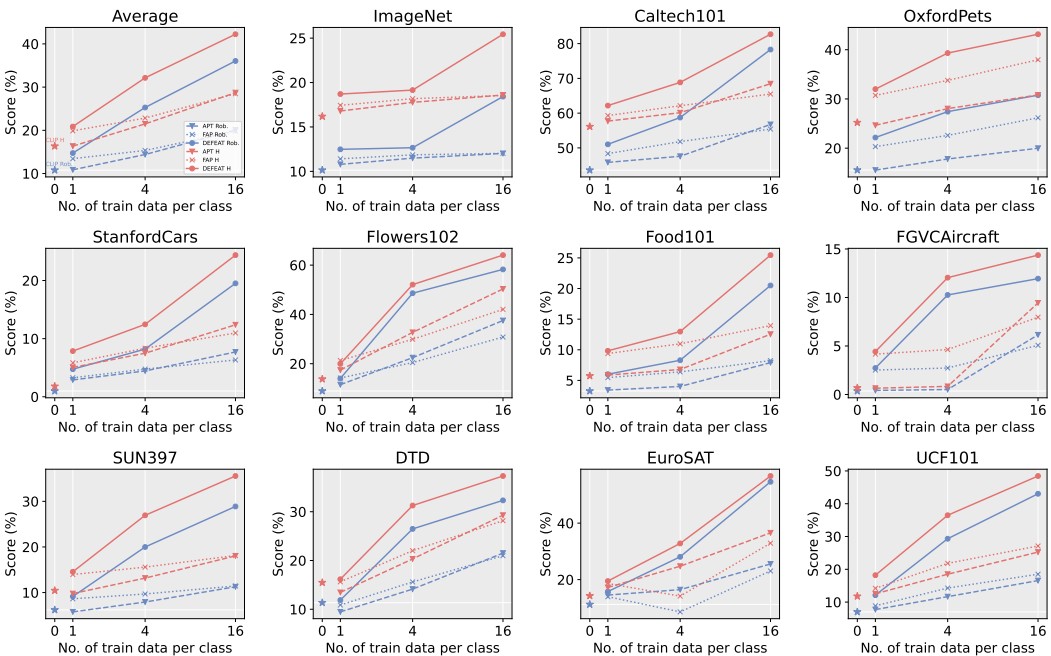

Figure A7: Performance on the 11 datasets under the adversarial few-shot classification setting when $\epsilon = 4/255$. As the accuracies among prompt tuning methods are comparable, we only plot the curves for robustness and 'H' to improve readability.

**Effect of $\lambda$.** In Eq. (10), $\lambda$ controls the weight of $\mathcal{L}_{\text{reg}}^{\text{I}}$, which is designed to ensure consistency between the $\hat{\mathbf{I}}_{\text{proj}}$ and the CLIP pre-trained class feature $\mathbf{I}$. According to Figure A8c, as $\lambda$ increases, the overall performance of DEFEAT improves, and then remains stable thereafter (i.e., $\lambda > 10$), making it easy to choose an appropriate $\lambda$ in practice. Those results highlight the importance of regularizing $\hat{\mathbf{I}}_{\text{proj}}$. In our experiments, we set $\lambda = 10$.

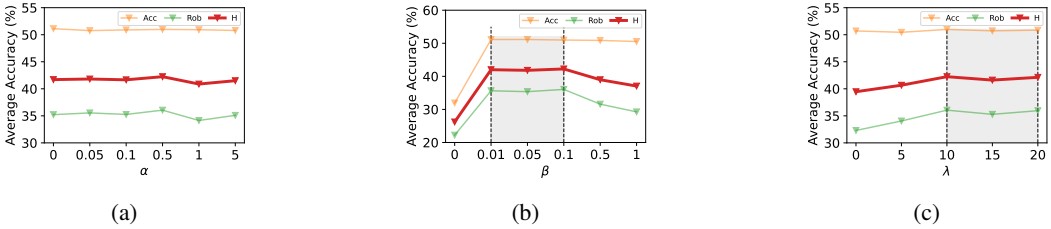

Figure A8: The average performance of DEFEAT on all datasets w.r.t. hyperparameters (i.e., $\alpha$, $\beta$, and $\lambda$) under the adversarial few-shot classification setting.

## C.3 ANALYSIS OF LOGITS FUSION STRATEGY

In this section, we analyse a variant of the logtis fusion strategy. Specifically, to control the trade-off between robustness and accuracy flexibility, we can modify the fused logits in Eq. (7) to the following form:

$$p(y|\mathbf{x}) = (1 - \omega)\, p^{\text{vq}}(y|\mathbf{x}) + \omega p^{\text{s}}(y|\mathbf{x}) \tag{11}$$

where $\omega$ is the hyperparameter. We analyze how different $\omega$ values impact DEFEAT's performance. Figure A9 shows that as $\omega$ increases, accuracy gradually improves because the logits of $\mathbf{I}$ and $\mathbf{t}^{\text{s}}$, which contribute to clean accuracy, have a larger weight in the fused logits. Robustness and 'H' initially increase with $\omega$, then plateau, and eventually decline. Although the range of $\omega$ values where 'H' stabilizes varies slightly across different datasets, $\omega = 0.5$ consistently falls within or near this range. Therefore, we set $\omega = 0.5$ for all datasets in our experiments.

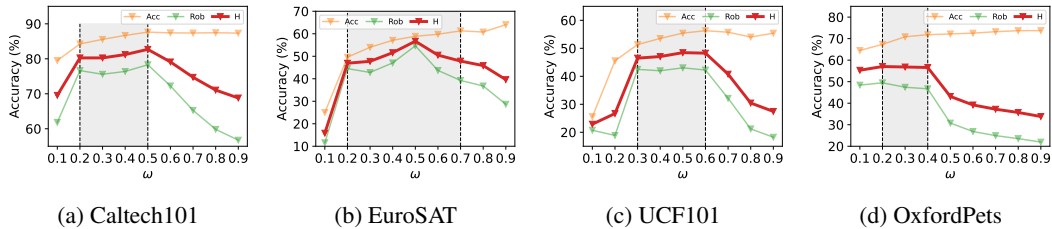

|   (a) Caltech101   |   (b) EuroSAT   |   (c) UCF101   |   (d) OxfordPets   |

Figure A9: Performance of DEFEAT on Caltech101, EuroSAT, UCF101, and OxfordPets datasets w.r.t. different $\omega$ values under the adversarial few-shot classification setting.

## C.4 ADVERSARIAL ROBUSTNESS EVALUATION UNDER VARIOUS ATTACKS

In this section, we evaluate the adversarial robustness of prompt tuning methods using a wider variety of attacks, including Carlini & Wagner (CW) attack (Carlini & Wagner, 2017) and AutoAttack (AA) (Croce & Hein, 2020). CW is a stronger optimization-based adversarial attack that finds the minimal perturbation required to cause a model to misclassify an input. AutoAttack (Croce & Hein, 2020) is a standardized benchmark suite that combines multiple attacks to evaluate a model's adversarial robustness reliably. Specifically, we use the standard AutoAttack setting, which executes a suite of four attacks: APGD-CE (untargeted), APGD-T (targeted), FAB-T (targeted), and Square (black-box). All component attacks utilize the default setting of 1 restart. We apply non-deterministic seeding to each component attack. As shown in Table A7, both CW and AA generate more potent adversarial examples than PGD, resulting in lower robustness for all methods (i.e., APT, FAP, and DEFEAT). Despite this, DEFEAT consistently outperforms the APT and FAP baselines across all datasets. This consistent superiority against a diverse and powerful suite of attacks provides strong evidence that DEFEAT's robustness is genuine and not an artifact of attack-specific overfitting or gradient obfuscation.

Table A7: Performance of adversarial prompt tuning methods under the adversarial few-shot classification setting with various attacks.

| 16shot, $\epsilon = 4/255$ | **Average** | | | | Caltech101 | | | | OxfordPets | | | | StanfordCars | | | | Flowers102 | | | |
|---|---|---|---|---|---|---|---|---|---|---|---|---|---|---|---|---|---|---|---|---|
| Method | Acc. | PGD | AA | CW | Acc. | PGD | AA | CW | Acc. | PGD | AA | CW | Acc. | PGD | AA | CW | Acc. | PGD | AA | CW |
| TeCoA | 33.00 | 11.37 | 9.64 | 11.07 | 78.78 | 43.61 | 40.93 | 44.30 | 66.26 | 15.56 | 11.28 | 15.26 | 10.32 | 0.99 | 0.62 | 0.99 | 30.13 | 8.93 | 6.54 | 7.59 |
| APT | 52.85 | 22.18 | 17.27 | 20.07 | 86.29 | 56.75 | 53.75 | 56.51 | 67.29 | 19.98 | 13.33 | 16.98 | 31.60 | 7.70 | 3.69 | 5.19 | **76.41** | 37.52 | 30.86 | 33.46 |
| FAP | **53.17** | 21.62 | 16.02 | 17.58 | 80.04 | 55.42 | 50.87 | 53.35 | 69.09 | 26.17 | 16.46 | 18.34 | **41.06** | 6.33 | 2.43 | 4.00 | 66.10 | 30.82 | 23.02 | 24.81 |
| DEFEAT | 52.60 | **39.92** | **27.02** | **34.89** | **87.63** | **78.30** | **70.30** | **77.32** | **72.17** | **30.77** | **18.86** | **23.63** | 32.45 | **19.51** | **9.63** | **16.71** | 71.13 | **58.26** | **45.11** | **55.38** |

| | Food101 | | | | FGVCAircraft | | | | DTD | | | | EuroSAT | | | | UCF101 | | | |
|---|---|---|---|---|---|---|---|---|---|---|---|---|---|---|---|---|---|---|---|---|
| Method | Acc. | PGD | AA | CW | Acc. | PGD | AA | CW | Acc. | PGD | AA | CW | Acc. | PGD | AA | CW | Acc. | PGD | AA | CW |
| TeCoA | 23.52 | 3.27 | 1.81 | 2.57 | 7.17 | 0.36 | 0.06 | 0.21 | 24.29 | 11.35 | 9.75 | 10.17 | 19.56 | 11.22 | 10.49 | 11.30 | 36.98 | 7.01 | 5.26 | 7.22 |
| APT | 30.39 | 7.90 | 4.48 | 5.54 | **20.31** | 6.15 | 3.06 | 4.05 | **45.86** | 21.51 | 15.07 | 17.14 | **64.33** | 25.54 | 18.35 | 25.04 | 53.16 | 16.55 | 12.82 | 16.68 |
| FAP | **45.60** | 8.23 | 3.92 | 5.37 | 18.57 | 5.07 | 2.58 | 3.06 | 42.85 | 20.98 | 17.32 | 18.09 | 64.51 | 23.06 | 15.19 | 16.33 | 50.67 | 18.50 | 12.42 | 14.83 |
| DEFEAT | 33.57 | **20.50** | **11.08** | **18.23** | 18.03 | **11.94** | **6.48** | **10.41** | 44.15 | **42.33** | **25.95** | **30.85** | 58.88 | **54.65** | **27.35** | **40.00** | 55.43 | **43.03** | **28.42** | **41.48** |

## C.5 ADAPTIVE ATTACK ON DEFEAT

To provide a more rigorous evaluation and to demonstrate DEFEAT's robustness, we design a defense-aware adaptive attack specifically tailored to circumvent its core architectural components. Following the principles outlined by Athalye et al. (2018), an adaptive attack must take the specific defense mechanism into account. Our attack targets the logits fusion strategy, which is a unique characteristic of the DEFEAT framework.

An attacker with knowledge of the DEFEAT architecture would realize that the final prediction comes from a standard, continuous feature branch and a defended, discretized feature branch. Based on this insight, we hypothesize that the most effective adaptive attack strategy is to identify the more vulnerable of these two branches and concentrate the full adversarial pressure on it.

To validate this hypothesis and identify the worst-case attack, we design a logits fusion-aware adaptive attack. Instead of maximizing the standard cross-entropy loss on the fused output, we maximize a weighted sum of the individual cross-entropy losses from each branch. The adaptive attack loss

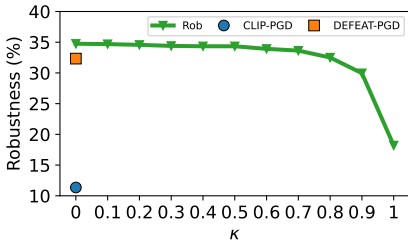

Figure A10: Robustness of DEFEAT on the DTD dataset under the logits fusion-aware adaptive attack with varying $\kappa$ values. Standard PGD attacks on CLIP and DEFEAT are provided as baselines.

$\mathcal{L}_{\text{adapt}}$ is defined as:

$$\mathcal{L}_{\text{ce}}^{\text{s}}(\mathbf{x}, \mathbf{t}^{\text{s}}, \mathbf{y}) = \sum_{i=1}^{C} y_i \log p^{\text{s}}(y|\mathbf{x}) \tag{12}$$

$$\mathcal{L}_{\text{ce}}^{\text{vq}}(\mathbf{x}, \mathbf{t}^{\text{s}}, \mathbf{y}) = \sum_{i=1}^{C} y_i \log p^{\text{vq}}(y|\mathbf{x}) \tag{13}$$

$$\mathcal{L}_{\text{adapt}} = \kappa * \mathcal{L}_{\text{ce}}^{\text{s}}(\mathbf{x}, \mathbf{t}^{\text{s}}, \mathbf{y}) + (1 - \kappa) * \mathcal{L}_{\text{ce}}^{\text{vq}}(\mathbf{x}, \mathbf{t}^{\text{s}}, \mathbf{y}) \tag{14}$$

where $p^{\text{s}}(y|\mathbf{x})$ and $p^{\text{vq}}(y|\mathbf{x})$ are defined in Eq. (4) and Eq. (6), $\kappa$ is a hyperparameter controls the focus of the attack. By varying the value of $\kappa$, we can explore the effect of attacking different branches. We then use a PGD-based optimizer to find the perturbation that maximizes $\mathcal{L}_{\text{adapt}}$.

We conduct experiments on the DTD dataset under the adversarial few-shot (16-shot) classification setting with $\epsilon = 4/255$ to analyze DEFEAT's performance under this adaptive attack. The results are presented in Figure A10.

The experimental results clearly validate our hypothesis. When we concentrate the full attack pressure on the standard, non-discretized feature branch ($\kappa = 1$), DEFEAT's robustness drops to its lowest point of 18.20%. This demonstrates that, with knowledge of DEFEAT's defense mechanism, the most effective adaptive attack is indeed to focus entirely on the branch unprotected by the PerturbShield module.

Conversely, when the attack exclusively targets the defended PerturbShield branch ($\kappa = 0$), the model's robustness reaches its highest point of 34.75%, even surpassing its performance against a standard PGD attack on the fused logits (32.33%). This provides strong evidence for the effectiveness of our proposed discretization module.

In summary, this experiment validates the design of our adaptive attack. Even under this custom-designed, worst-case adaptive attack, DEFEAT's robustness (18.20%) remains significantly higher than the CLIP baseline (11.35%), providing evidence for the good robustness of DEFEAT.

### C.6 GENERALIZATION TO STRONGER PERTURBATION BUDGETS

A critical measure of a defense mechanism's effectiveness is its ability to generalize to attacks stronger than those seen during training. To evaluate this, we conduct a challenging experiment where all models are trained using 16 examples per class under a standard PGD attack with a perturbation budget of $\epsilon = 4/255$, but are then evaluated against a much stronger, unseen PGD attack with $\epsilon = 8/255$.

As shown in Table A8, the results demonstrate a consistent advantage for DEFEAT. While the stronger attack significantly degrades the robustness of all baselines (compared to the 16-shot results in Table A6), DEFEAT maintains a substantially higher level of robustness across every dataset. On average, DEFEAT achieves a robust accuracy of 23.32%, which is over 4 times higher than FAP (5.81%) and 5 times higher than APT (4.62%).

Crucially, this significant gain in robustness is achieved with no sacrifice in clean accuracy. DEFEAT remains competitive with the baselines in this regard. This experiment provides compelling evidence that DEFEAT learns a more fundamental and generalizable defense mechanism. Instead of merely overfitting to a specific perturbation budget, its discrete feature reconstruction allows it to more

effectively ablate perturbations, making it a more reliable defense for real-world scenarios where attack strengths are unknown.

Table A8: Performance against larger perturbation budgets (trained on $\epsilon = 4/255$, tested on $\epsilon = 8/255$).

| train ($\epsilon = 4/255$) | | **Average** | | Caltech101 | | OxfordPets | | EuroSAT | | UCF101 | | DTD | | Flowers102 | | StanfordCars | | FGVCAircraft | |
| test ($\epsilon = 8/255$) | Method | Acc. | Rob. | Acc. | Rob. | Acc. | Rob. | Acc. | Rob. | Acc. | Rob. | Acc. | Rob. | Acc. | Rob. | Acc. | Rob. | Acc. | Rob. |
|---|---|---|---|---|---|---|---|---|---|---|---|---|---|---|---|---|---|---|---|
| | APT | 55.66 | 4.62 | 86.29 | 20.89 | 67.29 | 1.31 | 64.33 | 1.62 | 53.16 | 1.43 | 45.86 | 6.68 | 76.41 | 4.26 | 31.60 | 0.52 | 20.31 | 0.27 |
| PGD | FAP | 53.22 | 5.81 | 80.04 | 25.03 | 69.09 | 2.02 | 57.37 | 2.41 | 50.67 | 2.06 | 42.85 | 8.33 | 66.10 | 5.68 | 41.06 | 0.29 | 18.57 | 0.66 |
| | DEFEAT | 54.98 | **23.32** | 87.63 | **58.46** | 72.17 | **2.94** | 58.88 | **31.09** | 55.43 | **21.46** | 44.15 | **19.56** | 71.13 | **36.09** | 32.45 | **7.61** | 18.03 | **9.33** |

## C.7 GENERALIZATION TO ALTERNATIVE ROBUST BACKBONES

To demonstrate that the effectiveness of DEFEAT is not limited to a specific robust backbone, we replace the TeCoA-trained image encoder (Mao et al., 2023) with a more recent and powerful one pre-trained via the FARE method (Schlarmann et al., 2024). We then evaluate all adversarial prompt tuning baselines under the adversarial 16-shot classification setting.

The results in Table A9 show that DEFEAT maintains its superiority. Under the weaker PGD attack ($\epsilon = 1/255$), DEFEAT achieves an average robustness of 69.33%, outperforming the next-best baseline (FAP) by nearly 15%. This advantage becomes even more pronounced under the stronger attack ($\epsilon = 4/255$), where DEFEAT's average robustness of 52.46% is over 20% higher than the strongest baseline (FAP at 30.53%). Those results validate that DEFEAT is a generalizable framework that enhances robustness independently of the underlying pre-training method, and its benefits are magnified when paired with stronger robust encoders.

Table A9: Performance on a CLIP ViT-B/32 image encoder pre-trained with FARE (Schlarmann et al., 2024).

| $\epsilon = 1/255$ | | **Average** | | Caltech101 | | OxfordPets | | EuroSAT | | UCF101 | | DTD | | Flowers102 | | StanfordCars | | FGVCAircraft | |
| | Method | Acc. | Rob. | Acc. | Rob. | Acc. | Rob. | Acc. | Rob. | Acc. | Rob. | Acc. | Rob. | Acc. | Rob. | Acc. | Rob. | Acc. | Rob. |
|---|---|---|---|---|---|---|---|---|---|---|---|---|---|---|---|---|---|---|---|
| | FARE | 54.87 | 34.09 | 91.12 | 76.63 | 86.92 | 61.60 | 24.89 | 13.83 | 59.08 | 33.81 | 41.13 | 26.60 | 62.16 | 33.78 | 56.97 | 21.43 | 16.65 | 5.04 |
| | +APT | 75.57 | 50.60 | 95.25 | 84.06 | 87.87 | 63.29 | 79.58 | 42.44 | 78.91 | 51.86 | 63.95 | 41.31 | 93.10 | 73.08 | 72.89 | 37.71 | 32.97 | 11.07 |
| PGD | +FAP | 71.73 | 54.79 | 93.67 | 86.17 | 88.53 | 72.31 | 80.49 | 52.46 | 72.83 | 55.80 | 61.94 | 45.80 | 85.06 | 72.07 | 63.41 | 37.78 | 27.93 | 15.96 |
| | +DEFEAT | 74.75 | **69.33** | 95.17 | **93.02** | 89.78 | **83.02** | 76.63 | **68.88** | 79.57 | **74.23** | 64.13 | **58.39** | 91.64 | **87.09** | 70.71 | **63.11** | 30.33 | **26.91** |
| $\epsilon = 4/255$ | | **Average** | | Caltech101 | | OxfordPets | | EuroSAT | | UCF101 | | DTD | | Flowers102 | | StanfordCars | | FGVCAircraft | |
| | Method | Acc. | Rob. | Acc. | Rob. | Acc. | Rob. | Acc. | Rob. | Acc. | Rob. | Acc. | Rob. | Acc. | Rob. | Acc. | Rob. | Acc. | Rob. |
| | FARE | 42.84 | 15.87 | 86.00 | 54.69 | 77.60 | 24.64 | 16.28 | 10.88 | 43.01 | 8.96 | 31.26 | 15.19 | 39.22 | 7.75 | 39.17 | 3.56 | 10.20 | 1.32 |
| | +APT | 64.86 | 29.04 | 92.41 | 66.61 | 81.36 | 30.36 | 57.42 | 24.96 | 67.96 | 24.45 | 53.31 | 23.05 | 84.49 | 42.55 | 57.08 | 14.99 | 24.84 | 5.37 |
| PGD | +FAP | 58.59 | 30.53 | 85.48 | 66.41 | 77.41 | 39.22 | 59.53 | 23.72 | 59.19 | 26.38 | 46.99 | 25.18 | 70.44 | 41.41 | 47.62 | 13.52 | 22.05 | 8.43 |
| | +DEFEAT | 65.53 | **52.46** | 92.54 | **86.57** | 84.19 | **52.79** | 63.77 | **52.02** | 69.10 | **55.30** | 52.19 | **41.84** | 80.63 | **68.49** | 58.03 | **42.72** | 23.79 | **19.98** |

## C.8 GENERALIZATION TO ALTERNATIVE VLM BACKBONES

To demonstrate that the effectiveness of DEFEAT is not limited to the ViT-B/32 CLIP backbone used in our main paper, we conduct additional experiments on two distinct and powerful backbones: a larger CLIP model (ViT-L/14) and a model from a different pre-training family (EVA-CLIP (Sun et al., 2023)). The following results validate that DEFEAT is a broadly applicable framework.

First, to assess the scalability of our method, we replace the TeCoA-trained ViT-B/32 (Mao et al., 2023) backbone with the much larger TeCoA-trained ViT-L/14. As shown in Table A10, DEFEAT continues to provide significant robustness gains on this more powerful model. Those results demonstrate that our method scales effectively with model size.

Next, we evaluate DEFEAT's adaptability by applying it to EVA-CLIP (Sun et al., 2023), which represents a different pre-training paradigm compared to the original OpenAI CLIP. The results in Table A11 show that DEFEAT successfully enhances the robustness of this distinct model family. Those results validate that DEFEAT can serve as a general-purpose robust prompt tuning framework.

Table A10: Performance on the TeCoA-trained CLIP ViT-L/14 (Mao et al., 2023) backbone against PGD attack ($\epsilon = 4/255$) under the adversarial few-shot (16-shot) classification setting.

| $\epsilon = 4/255$ | Method | Caltech101 | | | DTD | | | Flowers102 | | | Average | | |
|---|---|---|---|---|---|---|---|---|---|---|---|---|---|
| | | Acc. | Rob. | H | Acc. | Rob. | H | Acc. | Rob. | H | Acc. | Rob. | H |
| PGD | CLIP | 94.81 | 17.85 | 30.04 | 53.72 | 0.59 | 1.17 | 79.46 | 0.45 | 0.89 | 76.00 | 6.30 | 11.63 |
| | TeCoA | 87.38 | 75.98 | 81.28 | 34.63 | 28.19 | 31.08 | 39.02 | 25.38 | 30.76 | 53.68 | 43.18 | 47.86 |
| | +APT | 95.42 | 86.77 | 90.89 | 57.21 | 44.92 | 50.33 | 88.67 | 76.53 | 82.15 | 80.43 | 69.41 | 74.51 |
| | +DEFEAT | 95.70 | **91.81** | **93.71** | 57.57 | **46.81** | **51.64** | 85.95 | **79.01** | 82.33 | 79.74 | **72.54** | **75.97** |

Table A11: Performance on the EVA-CLIP (Sun et al., 2023) backbone against PGD attack ($\epsilon = 1/255$) under the adversarial few-shot (16-shot) classification setting.

| $\epsilon = 1/255$ | Method | Caltech101 | | | DTD | | | Flowers102 | | | Average | | |
|---|---|---|---|---|---|---|---|---|---|---|---|---|---|
| | | Acc. | Rob. | H | Acc. | Rob. | H | Acc. | Rob. | H | Acc. | Rob. | H |
| PGD | EVA-CLIP | 97.20 | 3.81 | 7.33 | 49.41 | 2.78 | 5.26 | 75.92 | 0.81 | 1.60 | 74.18 | 2.47 | 4.77 |
| | +APT | 97.85 | 5.23 | 9.93 | 74.70 | 2.60 | 5.03 | 98.05 | 2.64 | 5.14 | 90.20 | 3.49 | 6.72 |
| | +DEFEAT | 96.80 | **11.70** | **20.88** | 71.28 | **5.61** | **10.40** | 95.86 | **4.47** | **8.54** | 87.98 | **7.26** | **13.41** |

## C.9 ROBUST VISUAL BACKBONE RELIANCE

In this section, we analyse the reliance of the robust visual backbone on the proposed DEFEAT method. While DEFEAT significantly enhances adversarial robustness under few-shot prompt tuning conditions, it relies heavily on pre-trained robust visual backbones (e.g., TeCoA (Mao et al., 2023)) like other adversarial prompt tuning methods (e.g., APT (Li et al., 2024) and FAP (Zhou et al., 2024)). The results in Table A12 show that APT failed to improve robustness when not using TeCoA. DEFEAT and FAP show less dependence on TeCoA when the perturbation size is small (i.e., $\epsilon = 1/255$). However, using the robust image encoder TeCoA still substantially enhances the robustness of both FAP and DEFEAT. When increasing perturbation size to $\epsilon = 4/255$, we find that all methods, i.e., APT, FAP, and DEFEAT) exhibit near-zero robustness or even collapse during training, a phenomenon also reported in Li et al. (2024); Zhou et al. (2024). The reasons may be the considerably smaller parameter space available for tuning compared to fine-tuning the entire model. Those results show that a robust visual backbone is necessary for adversarial prompt tuning methods.

Table A12: Performance of adversarial prompt tuning methods with (w/) or without (w/o) TeCoA on Flowers102 and DTD datasets for 16 shots under the adversarial few-shot classification setting when $\epsilon = 1/255$. The numbers in the parentheses denote the improvement or decline compared to the methods without TeCoA.

| Dataset | $\epsilon = 1/255$ | TeCoA | APT | | FAP | | DEFEAT | |
|---|---|---|---|---|---|---|---|---|
| | | | w/o TeCoA | w/ TeCoA | w/o TeCoA | w/ TeCoA | w/o TeCoA | w/ TeCoA |
| Flowers102 | Acc. | 48.15 | 85.06 (+36.91) | 86.80 (+38.65) | 80.15 (+32.00) | 82.46 (+34.31) | **91.72 (+43.57)** | 84.90 (+36.75) |
| | Rob. | 32.68 | 0.85 (-31.83) | 73.57 (+40.89) | 51.36 (+18.68) | 66.63 (+33.95) | 55.99 (+23.31) | **81.81 (+49.13)** |
| | H | 38.93 | 1.68 (-37.25) | 79.64 (+40.71) | 62.60 (+23.67) | 73.70 (+34.77) | 69.53 (+30.60) | **83.33 (+44.40)** |
| DTD | Acc. | 31.97 | 60.11 (+28.14) | 55.26 (+23.29) | 55.50 (+23.53) | 56.68 (+24.71) | **65.13 (+33.16)** | 57.15 (+25.18) |
| | Rob. | 23.88 | 2.60 (-21.28) | 39.54 (+15.66) | 28.67 (+4.79) | 41.67 (+17.79) | 30.91 (+7.03) | **52.66 (+28.78)** |
| | H | 27.34 | 4.98 (-22.35) | 46.10 (+18.76) | 37.81 (+10.47) | 48.03 (+20.69) | 41.92 (+14.58) | **54.81 (+27.47)** |

## C.10 COMPUTATIONAL COST ANALYSIS

To evaluate the practicality and efficiency of DEFEAT, we provide a comprehensive analysis of its computational overhead compared to APT and FAP. For a fair comparison, we adopt the same experimental settings as APT (e.g., training epochs, schedules, and data augmentation), while for FAP, we maintain its original implementation to ensure a faithful reproduction of its results. All experiments were conducted on the ImageNet dataset under the adversarial few-shot (16-shot) classification setting with $\epsilon = 4/255$.

The results in Table A13 show that DEFEAT achieves a highly favorable trade-off between computational cost and robustness.

**Training Memory.** The training memory required by DEFEAT (21240M) is nearly identical to that of APT (21204M) and significantly lower than that of FAP (37500M). This indicates that our PerturbShield module adds minimal GPU memory overhead.

**Training Time.** The training time of DEFEAT (19.21h) is negligibly longer than APT's (18.99h) and is less than half that of FAP (39.04h), demonstrating a clear efficiency advantage.

**Inference Time.** During inference, the per-image processing time for DEFEAT (3.1431ms) is only a minor increase of $\tilde{0}$.15ms over APT (2.9959ms), which is insignificant for practical applications.

In summary, the computational overhead introduced by our method is minimal during both training and inference. This slight cost is a highly favorable trade-off for the substantial improvement in robustness that DEFEAT provides over both APT and FAP, showing the practicality and efficiency of our method.

Table A13: Computational cost and performance comparison on ImageNet under the adversarial few-shot (16-shot) classification setting with $\epsilon = 4/255$.

| Method | Train.Memory | Train.time (h) | Infer.time (ms) | Acc. | Rob. | H |
|---|---|---|---|---|---|---|
| APT | 21204M | 18.99 | 2.9959 | 41.06 | 12.02 | 18.60 |
| DEFEAT (ours) | 21240M | 19.21 (+0.22) | 3.1431 | **41.09** | **18.42** | **25.44** |
| FAP | 37500M | 39.04 | 3.0020 | 40.45 | 12.03 | 18.54 |

## C.11 PERFORMANCE COMPARISON UNDER THE $\ell_2$ THREAT MODEL

We extend our experiments to the $\ell_2$ threat model, evaluating all methods against the PGD-$\ell_2$ ($\epsilon = 0.5$) attack across 10 downstream datasets.

As shown in Table A14, DEFEAT performs well under the $\ell_2$ threat model. Our method achieves an average robustness of 50.24%, surpassing both APT (47.34%) and FAP (47.07%). This result demonstrates that the robustness gains provided by DEFEAT are generalizable and not restricted to the $\ell_\infty$ norm.

Table A14: Performance Comparison under the adversarial few-shot (16-shot) classification setting with the $\ell_2$ threat model.

| PGD-$\ell_2$ | Average | | | Caltech101 | | OxfordPets | | StanfordCars | | Flowers102 | | Food101 | | FGVCAircraft | | SUN397 | | DTD | | EuroSAT | | UCF101 | |
|---|---|---|---|---|---|---|---|---|---|---|---|---|---|---|---|---|---|---|---|---|---|---|---|
| Method | Acc. | Rob.. | H | Acc. | Rob. | Acc. | Rob. | Acc. | Rob. | Acc. | Rob. | Acc. | Rob. | Acc. | Rob. | Acc. | Rob. | Acc. | Rob. | Acc. | Rob. | Acc. | Rob. |
| TeCoA | 33.03 | 29.17 | 30.98 | 78.78 | 74.40 | 66.26 | 59.14 | 10.32 | 7.80 | 30.13 | 26.59 | 23.52 | 19.34 | 7.17 | 5.07 | 33.24 | 28.13 | 24.29 | 22.51 | 19.56 | 17.73 | 36.98 | 31.03 |
| APT | 52.09 | 47.34 | 49.60 | 86.29 | 84.42 | 67.29 | 61.43 | 31.60 | 29.30 | **76.41** | 68.69 | 30.39 | 26.08 | **20.31** | 15.99 | 45.21 | 39.85 | **45.86** | 40.01 | **64.33** | 58.90 | 53.16 | 48.72 |
| FAP | 52.22 | 47.07 | 49.51 | 80.04 | 77.36 | 69.09 | 61.43 | **41.06** | **33.33** | 66.10 | 61.63 | **45.60** | **38.26** | 18.57 | 15.48 | 43.66 | 38.17 | 42.85 | 39.36 | 64.51 | 59.78 | 50.67 | 45.94 |
| **DEFEAT** | 51.97 | **50.24** | **51.09** | **87.63** | **86.41** | **72.17** | **67.10** | 32.45 | 30.93 | 71.13 | **69.43** | 33.57 | 31.99 | 18.03 | **16.65** | **46.26** | **44.14** | 44.15 | **42.61** | 58.88 | **58.85** | 55.43 | 54.27 |

## C.12 COMBINATION WITH TEST-TIME DEFENSE STRATEGY

To address the scenario facing strong attacks without a pre-trained robust backbone, we propose **DEFEAT-T**, a variant of the DEFEAT method. DEFEAT-T successfully adapts our core idea to the non-robust backbone scenario by combining it with a test-time defense strategy (Shu et al., 2022; Wang et al., 2025; Sheng et al., 2025).

The core of DEFEAT-T is to pivot our method from an "active defender" (via adversarial training) to a "passive purifier" (via clean training), combined with an efficient, optimization-free test-time ensemble. The mechanism is structured into two phases:

**Phase 1: Clean Training.** We first recognized that adversarial training on a non-robust backbone could be a source of the training collapse. Therefore, we discard adversarial training. Instead, we train our DEFEAT framework (including the learnable prompt and PerturbShield module) on a standard (non-robust) backbone using clean data only. In this phase, our PerturbShield module learns to reconstruct the clean feature distribution of the non-robust backbone.

**Phase 2: Test-Time Ensemble Defense.** At test time, we execute forward-pass ensemble defense, avoiding per-sample backpropagation or optimization, which are major bottlenecks for test-time

prompt tuning methods (Shu et al., 2022; Wang et al., 2025; Sheng et al., 2025). First, we generate $N^{\mathrm{aug}}$ (e.g., $N^{\mathrm{aug}}$=64) augmented views (e.g., random crop, random horizontal flip) of the test sample. Second, all $N^{\mathrm{aug}}$ views are fed through our model trained from clean data, to get $N^{\mathrm{aug}}$ fused logits by Eq. (7) as shown in Figure 4. Next, to obtain the final prediction, one can directly average these $N^{\mathrm{aug}}$ fused logits. Here, to have better performance, we perform a reliability-based weighted average. Specifically, we use the $N^{\mathrm{aug}}$ class features (which are collected alongside the grid features) to compute a similarity matrix. We then average the top-$k$ (e.g., $k$=20) values in each row to get a reliability score for each augmented view. Finally, we use these reliability scores as weights for a weighted average of the corresponding fused logits to obtain the final prediction.

The rationale for this design is our hypothesis that this view augmentation and weighted-average prediction could mitigate the impact of unseen adversarial perturbations, and that combining it with our DEFEAT framework provides a second stage of purification.

Table A15: Combination adversarial prompt tuning methods with test-time defense strategy under the adversarial few-shot classification setting.

| 16shot, $\epsilon = 4/255$ | Method | Average | | | Caltech101 | | OxfordPets | | Flowers102 | | DTD | |
|---|---|---|---|---|---|---|---|---|---|---|---|---|
| | | Acc. | Rob. | H | Acc. | Rob. | Acc. | Rob. | Acc. | Rob. | Acc. | Rob. |
| zero-shot | CLIP | 72.38 | 2.28 | 4.42 | 91.03 | 7.79 | 87.38 | 1.20 | 66.91 | 0.12 | 44.21 | 0.00 |
| | TeCoA | 49.87 | 19.86 | 28.41 | 78.78 | 43.61 | 66.26 | 15.56 | 30.13 | 8.93 | 24.29 | 11.35 |
| Test-time prompt tuning | R-TPT | 70.23 | 50.86 | 59.00 | 90.83 | 77.00 | 85.12 | 57.32 | 62.28 | 39.46 | 42.67 | 29.67 |
| Train-time prompt tuning | APT w/o TeCoA | 14.05 | 0.07 | 0.14 | 43.77 | 0.16 | 3.38 | 0.00 | 3.45 | 0.00 | 5.61 | 0.12 |
| | DEFEAT w/o TeCoA | 18.45 | 0.16 | 0.32 | 51.89 | 0.41 | 3.05 | 0.00 | 9.62 | 0.00 | 9.22 | 0.24 |
| | APT (w/ TeCoA) | 68.96 | 33.94 | 45.49 | 86.29 | 56.75 | 67.29 | 19.98 | 76.41 | 37.52 | 45.86 | 21.51 |
| | DEFEAT (w/ TeCoA) | 68.77 | 49.92 | 57.84 | 87.63 | 78.30 | 72.17 | 30.77 | 71.13 | 58.26 | 44.15 | 32.33 |
| Train-time prompt tuning + test-time defense | APT-T (w/o TeCoA) | 86.72 | 43.33 | 57.78 | 95.01 | 71.08 | 90.19 | 32.54 | **94.76** | 39.18 | 66.90 | 30.50 |
| | DEFEAT-T (w/o TeCoA) | **87.76** | **75.96** | **81.43** | **96.23** | **93.63** | **91.88** | **86.54** | 94.60 | **66.38** | **68.32** | **57.27** |

**Empirical Verification.** As shown in Table A15, DEFEAT-T (w/o TeCoA) achieves good average performance (87.76% Acc., 75.96% Rob., 81.43% H), surpassing test-time prompt tuning methods like R-TPT (70.23% Acc., 50.86% Rob., 59.00% H), and the standard DEFEAT (w/ TeCoA) (68.77% Acc., 49.92% Rob. , 57.84% H) even without a robust backbone.

Simultaneously, we also extend the baseline APT to APT-T (w/o TeCoA) (i.e., APT trained on clean data, evaluated with the same test-time ensemble but without the logits fusion strategy and the PerturbShield module). We find that APT-T (w/o TeCoA) (43.33% Rob.) achieves performance comparable to the original DEFEAT (w/ TeCoA) (49.92% Rob.). However, its robustness is still significantly weaker than our DEFEAT-T (75.96% Rob.). This comparison further demonstrates the effectiveness of our proposed logits fusion strategy and PerturbShield.

Crucially, our DEFEAT-T method has a decisive efficiency advantage over existing test-time prompt tuning methods (Shu et al., 2022; Wang et al., 2025; Sheng et al., 2025). Existing methods require per-sample optimization (batch size=1) involving backpropagation, making them slow. Our DEFEAT-T is an inference-only scheme with no gradient calculations, allowing for batch parallelism and much faster inference. However, even so, the time cost of DEFEAT-T's inference is still much higher than that of the original DEFEAT. As shown in Table A16, the inference time of DEFEAT-T is approximately 2x slower than that of DEFEAT. On the other hand, the training time for DEFEAT-T is much shorter, as it does not require adversarial training.

In summary, DEFEAT and DEFEAT-T can be adopted in different scenarios. The original DEFEAT (w/ TeCoA) is ideal for low-latency inference scenarios, whereas DEFEAT-T is a perfectly viable and highly robust alternative when a robust backbone is not available.

Table A16: Computational cost and performance comparison between DEFEAT, DEFEAT-T, and R-TPT (Sheng et al., 2025) on Caltech101 under the adversarial few-shot (16-shot) classification setting with $\epsilon = 4/255$.

| Method | Train.Time (s) | Infer.Memory | Infer.Time (s/per img) | Acc. | Rob.. | H |
|---|---|---|---|---|---|---|
| DEFEAT (w/ TeCoA) | 3597 | 8481M | **0.09** | 93.39 | 91.12 | 92.24 |
| DEFEAT-T (w/o TeCoA) | 1079 | 10420M | 0.19 | **96.19** | **95.58** | **95.88** |
| R-TPT | 0 | 5248M (batch=1) | 6.63 | 90.83 | 86.13 | 88.42 |

## C.13 THEORETICAL JUSTIFICATION

Here we provide a theoretical justification for why clean and adversarially perturbed features can be mapped to the same discrete representation within the VQ-VAE framework.

Let $\text{Enc}(\cdot)$ denote the composite visual encoder, which consists of the CLIP image encoder followed by the trainable VQ-VAE encoder. The clean latent features output by this composite encoder are denoted by $z_e^c = \text{Enc}(x) \in \mathbb{R}^{n \times d}$ (where $n$ is the number of latent positions after encoding, $d$ is the dimensionality of the latent embedding). Under an adversarial attack with perturbation $\delta$, the adversarial latent features become:

$$z_e^a = \text{Enc}(x + \delta) = z_e^c + \Delta z, \tag{15}$$

where $\Delta z \in \mathbb{R}^{n \times d}$ is the mapped perturbation in the latent space.

The core of VQ-VAE is to discretize continuous latent features through a codebook $\mathcal{E} = \{e_k\}_{k=1}^{K}$ (where $K$ is the codebook size). The covering radius $r$ of the codebook is defined as

$$r = \max_{z \in \mathcal{Z}} \min_{e_k \in \mathcal{E}} \|z - e_k\|_2, \tag{16}$$

where $\mathcal{Z}$ is the set of latent features output by the encoder from all clean images. The covering radius $r$ characterizes the "coverage capability" of the codebook over the clean feature space: the distance from any clean feature to its nearest codebook element does not exceed $r$, and $r$ is minimized through the codebook loss of VQ-VAE.

Based on the above definition, if the perturbation magnitude in the latent space satisfies $\|\Delta z\|_2 \leq r$, it is statistically highly probable that both the clean feature $z_e^c$ and the adversarial feature $z_e^a$ map to the same codebook element $e_k \in \mathcal{E}$. Notably, the training objective in DEFEAT encourages images that are more similar at the pixel level to be closer in feature space. So the feature shift $\Delta z$ induced by minor perturbations is typically much smaller than the codebook covering radius $r$, effectively mitigating such adversarial attacks during the quantization process.

**Empirical Verification.** To validate this, we conduct a statistical analysis on the latent space using the Caltech101 dataset (16-shot, $\epsilon = 4/255$). We measure two key metrics across all image patches:

- The average magnitude of adversarial perturbation per patch: $\|\Delta z\|_2 \approx \mathbf{126.0}$.
- The average distance between clean latent features and their assigned codebook vectors (serving as a proxy for the safety radius): $r \approx \mathbf{432.3}$.

The experimental results show that the covering radius is approximately 3.43 times larger than the perturbation magnitude (432.3 vs. 126.0). This substantial margin implies that the vast majority of adversarial perturbations are insufficient to push the latent features out of their original discrete region defined by their assigned code. Consequently, the clean and adversarial patches are quantized to the same discrete codes, confirming the effectiveness of our defense.

## C.14 STABILITY ANALYSIS WITH MULTI-SEED EVALUATION

To ensure the reproducibility of our results and confirm that the reported performance is not an artifact of a single run, we conduct a stability analysis by re-running our primary adversarial few-shot classification experiments. Specifically, we evaluate the models under the 16-shot setting with PGD attack ($\epsilon = 4/255$) using three random seeds (0, 1, and 2).

Table A17 compares our originally reported results (derived from Table 1 and Table A6) with the new averages calculated across three seeds. As observed, the multi-seed averages align closely with our reported single-seed results. Crucially, the robustness advantage of DEFEAT remains consistent and stable. For instance, the multi-seed average harmonic mean for DEFEAT is 41.66%, which still substantially surpasses the strongest baselines, APT (28.90%) and FAP (28.07%). This consistency confirms that our results are reproducible and statistically robust.

## C.15 STABILITY ANALYSIS AGAINST PROMPT VARIATIONS

As shown in Table A19, DEFEAT exhibits remarkable stability. The robustness fluctuates only slightly (between 31.09% and 32.33%) across different prompt variations.

Table A17: Comparison between originally reported results and averages over 3 seeds

| 16shot, $\epsilon = 4/255$ Method | Average (reported) | | | Average on 3 seeds | | |
|---|---|---|---|---|---|---|
| | Acc. | Rob. | H | Acc. | Rob. | H |
| TeCoA | 33.67 | 10.79 | 16.34 | 33.67 | 10.78 | 16.33 |
| APT | **51.08** | 20.26 | 29.02 | **51.27** | 20.12 | 28.90 |
| FAP | 50.50 | 19.82 | 28.47 | 50.68 | 19.41 | 28.07 |
| DEFEAT | 50.98 | **36.05** | **42.24** | 50.85 | **35.28** | **41.66** |

Table A18: Complete results calculated by averaging three seeds across all datasets. (16shot, $\epsilon = 4/255$)

| Method | ImageNet | | Caltech101 | | OxfordPets | | StanfordCars | | Flowers102 | | Food101 | | FGVCAircraft | | SUN397 | | DTD | | EuroSAT | | UCF101 | |
|---|---|---|---|---|---|---|---|---|---|---|---|---|---|---|---|---|---|---|---|---|---|---|
| | Acc. | Rob. | Acc. | Rob. | Acc. | Rob. | Acc. | Rob. | Acc. | Rob. | Acc. | Rob. | Acc. | Rob. | Acc. | Rob. | Acc. | Rob. | Acc. | Rob. | Acc. | Rob. |
| TeCoA | 40.11 | 10.15 | 78.78 | 43.57 | 66.26 | 15.49 | 10.32 | 0.99 | 30.13 | 8.94 | 23.52 | 3.26 | 7.17 | 0.36 | 33.24 | 6.21 | 24.29 | 11.39 | 19.56 | 11.21 | 36.98 | 6.99 |
| APT | **40.98** | 12.22 | 86.38 | 56.79 | 68.25 | 19.82 | 33.08 | 7.74 | **76.24** | 37.18 | 30.19 | 7.87 | **19.50** | 6.04 | 45.32 | 11.28 | 44.09 | 20.06 | **65.34** | 25.87 | 54.55 | 16.50 |
| FAP | 40.53 | 12.10 | 80.36 | 55.50 | 69.27 | 25.59 | **40.91** | 6.60 | 66.11 | 30.96 | **45.18** | 8.30 | 19.00 | 4.97 | 43.75 | 11.33 | 43.31 | 17.71 | 58.14 | 22.64 | 50.87 | 17.82 |
| DEFEAT | 40.79 | **18.02** | **86.96** | **77.61** | **72.12** | **32.31** | 32.94 | **20.38** | 71.00 | **58.13** | 33.60 | **20.50** | 16.82 | **11.32** | **46.23** | **29.61** | **44.41** | **32.57** | 58.85 | **45.35** | **55.59** | **42.34** |

Crucially, the performance trends remain consistent across all variations. Even under the least optimal prompt setting ("{} texture."), DEFEAT (31.09% Rob.) significantly outperforms the strongest baseline, APT (20.26% Rob.), by a margin of +10.83%. This confirms that our method's superiority is intrinsic to its design rather than an artifact of specific prompt engineering.

Table A19: Stability analysis of DEFEAT against different hand-crafted prompt templates on the DTD dataset.

| 16shot, $\epsilon = 4/255$ | DTD | | |
|---|---|---|---|
| Method | Acc. | Rob. | H |
| TeCoA | 33.67 | 10.79 | 16.34 |
| APT | 51.08 | 20.26 | 29.02 |
| FAP | 50.50 | 19.82 | 28.47 |
| **DEFEAT with different hand-crafted prompts** | | | |
| "a photo of a {}, a type of texture." | 44.15 | 32.33 | 37.33 |
| "{} texture." | 44.74 | 31.09 | 36.69 |
| "a photo of {}." | 44.03 | 31.80 | 36.93 |

# D USE OF LARGE LANGUAGE MODELS

The LLM served as an assistive tool for language editing and manuscript polishing. Its usage was confined to improving grammar, refining phrasing for clarity, and ensuring idiomatic English.

