# OpenReview forum: "Discrete Latent Features Ablate Adversarial Attack: A Robust Prompt Tuning Framework for VLMs"
_ICLR.cc/2026/Conference — ICLR 2026 Poster_

### Official Review · Reviewer_DTmv · 2025-10-20

**Soundness:** 3
**Presentation:** 3
**Contribution:** 3
**Rating:** 6
**Confidence:** 3

**Summary:**

The paper proposes DEFEAT, a robust prompt-tuning framework for CLIP-like VLMs that discretizes latent grid features with a VQ-VAE module, and fuses logits from discretized and original branches, coupled with prompt-alignment regularization to hand-crafted prompts. The central claim is that feature discretization reduces clean->adv feature shift, yielding substantially higher robust accuracy and harmonic mean scores under few-shot, cross-dataset, and domain-generalization settings. The paper provides PGD evaluations, AutoAttack and CW, and a defense-aware adaptive attack targeting the fusion branches. Overall, the method is compact (frozen encoders; train *PerturbShield* + prompts) and the empirical coverage is broad, though statistical reporting and adaptive-attack rigor need tightening.

**Strengths:**

- Clear discretization motivation with reduced feature shift
- Simple modular defense (VQ-VAE grid reconstruction + projection + fusion + prompt regularization)
- Broad empirical coverage with consistent robustness/H gains; AA/CW and adaptive checks provided
- Ablations clarify each component’s role; µ improves robustness

**Weaknesses:**

- Adaptive attack lacks explicit BPDA/ST gradients through quantization, risking masking
- AA/CW only on subset; AA configs (targeting, restarts, seeds) unspecified in main text
- Threat model focused on ℓ∞; no ℓ2/spatial in main paper
- No multi-seed CIs; missing runtime and codebook specs

**Questions:**

- What are the typical failure cases where discretization/fusion underperform (e.g., fine-grained textures, clutter, small objects), and what mitigation directions look promising?
- Are the reported robustness gains consistent across multiple seeds, and do they remain visible under paired comparisons against the strongest baseline?
- How stable are results to reasonable prompt variations (e.g., templated vs hand-crafted phrasing), and do trends remain when prompts are slightly perturbed?

---

> ### Author Response · Authors · 2025-11-22
> **Response by Authors, Part 1**
>
> Thank you for your constructive comments. Below we have made responses to your comments. If you have any further comment, please feel free to let us know and we are more than glad to discuss with you.
>
> ---
>
> > W1. Adaptive attack lacks explicit BPDA/ST gradients through quantization, risking masking.
>
> ### Answer to W1:
>
> Thanks for your comment. We respectfully clarify that our attack generation explicitly employs the **Straight-Through Estimator (STE)** inherent in the VQ-VAE implementation, which functions effectively as a Backward Pass Differentiable Approximation (BPDA) by copying gradients from the decoder input to the encoder output as noted in the original submission (Lines 162-164). The forward and backward propagation through quantization are as follows:
>
> * **Forward Pass**: Continuous features are mapped to the nearest discrete codebook vectors.
> * **Backward Pass**: Gradients are copied directly from the decoder input to the encoder output. This treats the non-differentiable quantization as an identity function during backpropagation, which provides the gradient approximation required for attacks.
>
> Furthermore, our evaluation using AutoAttack (Appendix C.4) includes the Square Attack, a gradient-free black-box method. DEFEAT consistently outperforms the APT and FAP baselines. This consistent superiority against a diverse and powerful suite of attacks provides strong evidence that DEFEAT’s robustness is not gradient obfuscation. Moreover, our logits fusion-aware adaptive attack successfully identified the worst-case scenario (attacking the unprotected branch, $\kappa=1$), significantly reducing the robustness to the defended branch or the fused branch. Therefore, we think our adaptive attack is rigorous.
>
> ---
>
> > W2. AA/CW only on subset; AA configs (targeting, restarts, seeds) unspecified in main text.
>
> ### Answer to W2:
>
> Thanks for your feedback. As suggested, **we now provide a more comprehensive evaluation against PGD, AutoAttack(AA), and CW attacks across 10 downstream datasets** rather than only across 3 datasets. We exclude ImageNet solely due to prohibitive computational costs (requiring tens of days on our available resources) within the rebuttal time.
>
> As shown in the tables below, DEFEAT's robustness advantage is consistent and significant against stronger attacks. On average, **DEFEAT** achieves **26.14%/34.14% (AA/CW)** robustness, substantially outperforming **APT (16.26%/19.00%)** and **FAP (14.82%/16.69%)**. **We have updated Table A7 in Appendix C.4.**
>
> |16shot, $\epsilon=4/255$|Average|||||
> |:---|:---:|:---:|:---:|:---:|:---:|
> |**Method**|**Acc.**|**PGD**|**AA**|**CW**|**H**|
> |TeCoA|33.03|10.85|9.09|10.55|12.22|
> |APT|52.09|21.09|16.26|19.00|22.13|
> |FAP|52.22|20.60|14.82|16.69|20.50|
> |DEFEAT|51.97|**38.82**|**26.14**|**34.14**|**35.54**|
>
> ||Cal.||||Pets||||Cars||||Flo.||||Food||||
> |:---|:---:|:---:|:---:|:---:|:---:|:---:|:---:|:---:|:---:|:---:|:---:|:---:|:---:|:---:|:---:|:---:|:---:|:---:|:---:|:---:|
> |**Method**|Acc.|PGD|AA|CW|Acc.|PGD|AA|CW|Acc.|PGD|AA|CW|Acc.|PGD|AA|CW|Acc.|PGD|AA|CW|
> |TeCoA|78.78|43.61|40.93|44.30|66.26|15.56|11.28|15.26|10.32|0.99|0.62|0.99|30.13|8.93|6.54|7.59|23.52|3.27|1.81|2.57|
> |APT|86.29|56.75|53.75|56.51|67.29|19.98|13.33|16.98|31.60|7.70|3.69|5.19|76.41|37.52|30.86|33.46|30.39|7.90|4.48|5.54|
> |FAP|80.04|55.42|50.87|53.35|69.09|26.17|16.46|18.34|41.06|6.33|2.43|4.00|66.10|30.82|23.02|24.81|45.60|8.23|3.92|5.37|
> |DEFEAT|87.63|78.30|70.30|77.32|72.17|30.77|18.86|23.63|32.45|19.51|9.63|16.71|71.13|58.26|45.11|55.38|33.57|20.50|11.08|18.23|
>
> ||Air.||||SUN||||DTD||||SAT||||UCF||||
> |:---|:---:|:---:|:---:|:---:|:---:|:---:|:---:|:---:|:---:|:---:|:---:|:---:|:---:|:---:|:---:|:---:|:---:|:---:|:---:|:---:|
> |**Method**|Acc.|PGD|AA|CW|Acc.|PGD|AA|CW|Acc.|PGD|AA|CW|Acc.|PGD|AA|CW|Acc.|PGD|AA|CW|
> |TeCoA|7.17|0.36|0.06|0.21|33.24|6.20|4.15|5.87|24.29|11.35|9.75|10.17|19.56|11.22|10.49|11.30|36.98|7.01|5.26|7.22|
> |APT|20.31|6.15|3.06|4.05|45.21|11.27|7.14|9.41|45.86|21.51|15.07|17.14|64.33|25.54|18.35|25.04|53.16|16.55|12.82|16.68|
> |FAP|18.57|5.07|2.58|3.06|43.66|11.44|3.98|8.68|42.85|20.98|17.32|18.09|64.51|23.06|15.19|16.33|50.67|18.50|12.42|14.83|
> |DEFEAT|18.03|11.94|6.48|10.41|46.26|28.88|18.20|27.43|44.15|42.33|25.95|30.85|58.88|54.65|27.35|40.00|55.43|43.03|28.42|41.48|
>
> As for **AA configs**, we apologize for not specifying these details in the main text and **have added them to Appendix C.4 (highlighted in blue).** As shown in our code of the Supplementary Material, we use the standard configuration from the official AutoAttack library:
> ```
> attack = AutoAttack(model, norm='Linf', eps=eps, version='standard', verbose=False)
> ```
> The `version='standard'` setting executes a diverse suite of four attacks: APGD-CE (untargeted), APGD-T (targeted), FAB-T (targeted), and Square (black-box). All component attacks utilize the default setting of 1 restart. We use the default `seed=None`, which applies non-deterministic seeding to each component attack.

---

> ### Author Response · Authors · 2025-11-22
> **Response by Authors, Part 2**
>
> > W3. Threat model focused on ℓ∞; no ℓ2/spatial in main paper.
>
> ### Answer to W3:
>
> Thanks for valuable suggestion. We have extended our experiments to the $\ell_2$ threat model, evaluating all methods against the PGD-L2 attack ($\epsilon=0.5$) across 10 downstream datasets.
>
> As shown in the table below, DEFEAT performs well under the $\ell_2$ threat model. Our method achieves an average robustness of 50.24%, surpassing both APT (47.34%) and FAP (47.07%). This result demonstrates that the robustness gains provided by DEFEAT are generalizable and not restricted to the $\ell_{\infty}$ norm. **We have included those experiments in Appendix C.11 (highlighted in blue).**
>
> |PGD-L2, $\epsilon=0.5$|Avg.|||Cal.||Pets||Cars||Flo.||Food||Air.||SUN||DTD||SAT||UCF||
> |:---|:---:|:---:|:---:|:---:|:---:|:---:|:---:|:---:|:---:|:---:|:---:|:---:|:---:|:---:|:---:|:---:|:---:|:---:|:---:|:---:|:---:|:---:|:---:|
> |**Method**|**Acc.**|**Rob.**|**H**|Acc.|Rob.|Acc.|Rob.|Acc.|Rob.|Acc.|Rob.|Acc.|Rob.|Acc.|Rob.|Acc.|Rob.|Acc.|Rob.|Acc.|Rob.|Acc.|Rob.|
> |TeCoA|33.03|29.17|30.98|78.78|74.40|66.26|59.14|10.32|7.80|30.13|26.59|23.52|19.34|7.17|5.07|33.24|28.13|24.29|22.51|19.56|17.73|36.98|31.03|
> |APT|52.09|47.34|49.60|86.29|84.42|67.29|61.43|31.60|29.30|76.41|68.69|30.39|26.08|20.31|15.99|45.21|39.85|45.86|40.01|64.33|58.90|53.16|48.72|
> |FAP|**52.22**|47.07|49.51|80.04|77.36|69.09|61.43|41.06|**33.33**|66.10|61.63|45.60|**38.26**|18.57|15.48|43.66|38.17|42.85|39.36|64.51|59.78|50.67|45.94|
> |**DEFEAT**|51.97|**50.24**|**51.09**|87.63|**86.41**|72.17|**67.10**|32.45|30.93|71.13|**69.43**|33.57|31.99|18.03|**16.65**|46.26|**44.14**|44.15|**42.61**|58.88|**58.85**|55.43|**54.27**|
>
> ---
>
> > W4. No multi-seed CIs; missing runtime and codebook specs.
>
> ### Answer to W4:
>
> Thanks for your comments. We address your concerns as follows:
>
> 1. **Multi-seed CIs**. We have now re-run our primary adversarial few-shot classification experiments (16-shot, PGD, $\epsilon=4/255$) using three seeds (0, 1, and 2). The following table compares our originally reported results (from Table 1 and Table A6) with the new, robust averages across three seeds.
>
> As shown, the multi-seed averages align closely with our reported results. Most importantly, DEFEAT's significant robustness advantage remains consistent and stable. (e.g., DEFEAT's new average 'H' is 41.66%, still far surpassing APT's 28.90% and FAP's 28.07%). This confirms that our results are reproducible and not an artifact of a single run. **We have included those experiments in Appendix C.14 (highlighted in blue).**
>
> | 16shot, $\epsilon=4/255$|Average (reported)|||Average on 3 seeds|||
> |:---|:---:|:---:|:---:|:---:|:---:|:---:|
> |**Method**|**Acc.**|**Rob.**|**H**|**Acc.**|**Rob.**|**H**|
> |TeCoA|33.67|10.79|16.34|33.67|10.78|16.33|
> |APT|**51.08**|20.26|29.02|**51.27**|20.12|28.90|
> |FAP|50.50|19.82|28.47|50.68|19.41|28.07|
> |DEFEAT|50.98|**36.05**|**42.24**|50.85|**35.28**|**41.66**|
>
> The following table shows the complete results, calculated by averaging three seeds across all datasets.
>
> |16shot,PGD, $\epsilon=4/255$|ImgNet||Cal.||Pets||Cars||Flo.||Food||Air.||SUN||DTD||SAT||UCF||
> |:---|:---:|:---:|:---:|:---:|:---:|:---:|:---:|:---:|:---:|:---:|:---:|:---:|:---:|:---:|:---:|:---:|:---:|:---:|:---:|:---:|:---:|:---:|
> |Method|Acc.|Rob.|Acc.|Rob.|Acc.|Rob.|Acc.|Rob.|Acc.|Rob.|Acc.|Rob.|Acc.|Rob.|Acc.|Rob.|Acc.|Rob.|Acc.|Rob.|Acc.|Rob.|
> |TeCoA|40.11|10.15|78.78|43.57|66.26|15.49|10.32|0.99|30.13|8.94|23.52|3.26|7.17|0.36|33.24|6.21|24.29|11.39|19.56|11.21|36.98|6.99|
> |APT|**40.98**|12.22|86.38|56.79|68.25|19.82|33.08|7.74|**76.24**|37.18|30.19|7.87|**19.50**|6.04|45.32|11.28|44.09|20.06|**65.34**|25.87|54.55|16.50|
> |FAP|40.53|12.10|80.36|55.50|69.27|25.59|**40.91**|6.60|66.11|30.96|**45.18**|8.30|19.00|4.97|43.75|11.33|43.31|17.71|58.14|22.64|50.87|17.82|
>
> 2. **Missing runtime**. We would like to clarify that a detailed analysis of the **computational cost was indeed included in Appendix C.10 (Table A13)** as noted in the original submission (Lines 472-474), which may have been overlooked. This section explicitly compares DEFEAT against baselines in terms of training memory, training time, and inference time. The results demonstrate that the overhead introduced by our method is minimal, confirming its practicality.
>
> 3. **Missing Codebook Specs**. We apologize for not specifying codebook specifications in the main text. **We have included these details in Appendix B (highlighted in blue).** As defined in our provided code, our PerturbShield module, which contains the VQ-VAE, is configured with specific parameters.
>     * Input Dimension: 512 (matching CLIP ViT-B/32 grid features).
>     * Latent Dimension: 256.
>     * Codebook Size: 512 unique embedding vectors.
>     * Architecture: The encoder comprises two convolutional layers and a residual block; the decoder mirrors this structure with a residual block and a transposed convolutional layer for reconstruction.

---

> ### Author Response · Authors · 2025-11-22
> **Response by Authors, Part 3**
>
> > Q1. What are the typical failure cases where discretization/fusion underperform (e.g., fine-grained textures, clutter, small objects), and what mitigation directions look promising?
>
> ### Answer to Q1:
>
> Thanks for your question. Here we provide a comprehensive analysis for failure cases in DEFEAT.
>
> 1. **Fine-Grained Features and Small Objects**.
>     * Failure: For tasks relying on fine-grained textures (e.g., DTD) or subtle object details (e.g., FGVCAircraft, OxfordPets), the VQ-VAE may "snap" distinct-but-similar features to the same codebook vector. This quantization-induced information loss could hinder the model's discriminative capability, which explains why our performance gains on these specific datasets (Table A6) are more modest compared to baselines.
>
>     * Mitigation: This is an inherent trade-off of this quantization level. A promising (but computationally expensive) direction would be to use a larger codebook or a hierarchical VQ-VAE to capture finer details.
>
> 2. **Data Scarcity**.
>     * Failure: Our logits fusion strategy requires both the learnable prompt and the PerturbShield module to be well-trained. When training data is extremely limited (e.g., **1-shot**), the PerturbShield's VQ-VAE and projection layer may be under-trained, leading to unsatisfactory feature reconstruction. This explains why our 1-shot performance is only comparable to the prior SOTA (i.e., FAP).
>
>     * Mitigation: This limitation is effectively resolved by using more data. As shown in Table A6, when given 4-shot and 16-shot data, the PerturbShield is trained effectively, and our method's performance significantly and consistently outperforms all baselines.
>
> 3. **Robust Backbone Reliance**.
>     * Failure: As noted in our paper, our adversarial training paradigm (like all existing adversarial prompt tuning methods) relies on a robust backbone (e.g., TeCoA). When using a standard, non-robust backbone, the training process could collapse under strong attacks.
>
>     * Mitigation: The solution is to combine our method with a test-time defense strategy. We have developed and tested a new variant, DEFEAT-T, which is designed specifically for this scenario. DEFEAT-T involves training our PerturbShield on clean data (avoiding training collapse) and then, at inference, applying a test-time ensemble defense. As our new experimental results show, DEFEAT-T (w/o TeCoA) achieves outstanding robustness (75.96% Rob.), even surpassing the original DEFEAT (w/ TeCoA). We have provided a full analysis of this DEFEAT-T method, including the method, results, and computational cost, in **Answer to Q1** for Reviewer QZPj regarding this specific topic.
>
> ---
>
> > Q2. Are the reported robustness gains consistent across multiple seeds, and do they remain visible under paired comparisons against the strongest baseline?
>
> ### Answer to Q2:
>
> Thanks for your question. This question is directly related to W4 ("No multi-seed CIs..."), which we have addressed in detail in our **Answer to W4**. To summarize:
>
> 1. **Consistency**: Yes, **the robustness gains are consistent across multiple seeds.** We have re-run our key experiments (16-shot, PGD, $\epsilon=4/255$) using three seeds (0, 1, and 2).
>
> 2. **Paired Comparison**: Yes, **the gains remain visible and significant**. As the new multi-seed average table shows, the average 'H' for DEFEAT is **41.66%**, which remains significantly higher than the baselines, **APT (28.90%) and FAP (28.07%**).
>
> Please refer to our **Answer to W4** for the detailed multi-seed results table.
>
>
> ---
>
> > Q3. How stable are results to reasonable prompt variations (e.g., templated vs hand-crafted phrasing), and do trends remain when prompts are slightly perturbed?
>
> ### Answer to Q3:
>
> Thanks for your insightful question. To empirically verify the stability of DEFEAT against prompt variations, we evaluate DEFEAT using three distinct hand-crafted prompt templates for regularization ($\mathcal{L}_{reg}^{T}$).
>
> As shown in the table below, DEFEAT exhibits remarkable stability. The robustness fluctuates only slightly (between 31.09% and 32.33%) across different prompt variations.
>
> Crucially, **the performance trends remain consistent across all variations.** Even under the least optimal prompt setting, DEFEAT (31.09% Rob.) significantly outperforms the strongest baseline, APT (20.26% Rob.), by a margin of +10.83%. This confirms that our method's superiority is intrinsic to its design rather than an artifact of specific prompt engineering. **We have included those experiments in Appendix C.15 (highlighted in blue).**
>
>
> |16shot, $\epsilon=4/255$|**DTD**|||
> |:---|:---:|:---:|:---:|
> |**Method**|**Acc.**|**Rob.**|**H**|
> |TeCoA|33.67|10.79|16.34|
> |APT|51.08|20.26|29.02|
> |FAP|50.50|19.82|28.47|
> |
> |**DEFEAT with different hand-crafted prompts**||||
> |"a photo of a {}, a type of texture."|44.15|32.33|37.33|
> |"{} texture."|44.74|31.09|36.69|
> |"a photo of {}."|44.03|31.80|36.93|

---

### Official Review · Reviewer_vWTM · 2025-10-27

**Soundness:** 3
**Presentation:** 3
**Contribution:** 2
**Rating:** 4
**Confidence:** 4

**Summary:**

This paper presents a novel and well-executed framework for improving the adversarial robustness of VLMs through prompt tuning. Extensive evaluation on 15 datasets under few-shot, cross-dataset and domain-generalisation settings.

**Strengths:**

1. Introduces a novel application of latent feature discretization to adversarial defense for VLMs.

2. Comprehensive evaluation across 15 datasets, covering adversarial few-shot classification, cross-dataset, and domain generalization.

**Weaknesses:**

1. This paper presents a well-structured framework. however, the underlying techniques for the individual components are not entirely novel.

2. The paper makes a significant contribution to improving the adversarial robustness of CLIP. However, a limitation of the current study is that all experiments are exclusively conducted on the CLIP model. To broaden the applicability and contribution of the proposed methods, it would be highly beneficial to include experimental results on other relevant models, such as OpenCLIP and EVA-CLIP. Furthermore, the empirical evaluation focuses exclusively on image classification.The absence of experiments on other applications limits the demonstrated scope and real-world applicability of the DEFEAT method.

3. The practical utility of the DEFEAT method is not fully clear due to the lack of analysis on its computational cost relative to baseline methods.

4. The discussion could be further enriched by considering recent advancements in test-time defenses. if the authors could investigate whether combining DEFEAT with pre-processing defenses [1] or test-time defenses [2,3] could lead to synergistic effects, potentially achieving a "1+1>2" outcome.

[1] Diffusion Models for Adversarial Purification, ICML 2022

[2] CLIP is Strong Enough to Fight Back: Test-time Counterattacks towards Zero-shot Adversarial Robustness of CLIP, CVPR 2025.

[3] TAPT: Test-Time Adversarial Prompt Tuning for Robust Inference in Vision-Language Models, CVPR 2025.

**Questions:**

Please refer to the questions raised in the Weaknesses section above.

---

> ### Author Response · Authors · 2025-11-22
> **Response by Authors, Part 1**
>
> Thank you for your constructive comments. Below we have made responses to your comments. If you have any further comment, please feel free to let us know and we are more than glad to discuss with you.
>
> ---
>
> > W1. This paper presents a well-structured framework. However, the underlying techniques for the individual components are not entirely novel.
>
> ### Answer to W1:
>
> Thanks for your feedback. While we agree that the establishment of DEFEAT is built on some existing techniques (e.g., VQ-VAE), we emphasize that our core contribution lies in the **novel perspective and non-trivial architectural design** tailored for VLM adversarial defense. This novelty has also been recognized by Reviewers QZpj and spep.
>
> Specifically, the novelty of DEFEAT includes:
>
> * **A New Defense Direction:** To the best of our knowledge, DEFEAT is the first work to systematically validate the effectiveness of **latent feature discretization** in mitigating adversarial attacks for VLMs. We introduce a new and unexplored direction for adversarial defense in this domain.
>
> * **Non-trivial Integration:** The integration of these components is far from a trivial concatenation. As shown in Fig. 3b, a naive application of VQ-VAE to input images causes catastrophic performance degradation. In contrast, DEFEAT avoids such degradation by dedicated design such as applying discretization to the grid features and designing a semantic alignment projection to ensure compatibility with CLIP's embedding space. We further establish a better trade-off between robustness and accuracy by using a logits fusion strategy to mitigate the impact of the information loss. Hence, our method is not a trivial combination of existing methods.
>
> * **High Effectiveness.** The effectiveness of this well-integrated framework is validated by substantial performance gains. As shown in Table 1, our method surpasses state-of-the-art methods by a large margin. For example, with 16-shot training under $\epsilon = 4/255$, DEFEAT surpasses APT (FAP) by 15.79% (16.23%) in terms of robustness and 13.22% (13.76%) in terms of ‘H’, respectively.
>
> In summary, while DEFEAT leverages existing techniques, it contributes a novel and solid framework to VLM robustness, and achieved state-of-the-art results.
>
> ---
>
> > W2. The paper makes a significant contribution to improving the adversarial robustness of CLIP. However, a limitation of the current study is that all experiments are exclusively conducted on the CLIP model. To broaden the applicability and contribution of the proposed methods, it would be highly beneficial to include experimental results on other relevant models, such as OpenCLIP and EVA-CLIP. Furthermore, the empirical evaluation focuses exclusively on image classification. The absence of experiments on other applications limits the demonstrated scope and real-world applicability of the DEFEAT method.
>
> ### Answer to W2:
>
> Thanks for your suggestion. We address your concerns as follows.
>
> 1. **Generalization to other VLM backbones.**
>
> **In fact, we have indeed extended DEFEAT to other relevant models (i.e., FARE-trained CLIP ViT-B/32 [9], TeCoA-trained CLIP ViT-L/14, and EVA-CLIP) in Appendix (as noted in the original submission (Lines 472-474)), which may have been overlooked.** Specifically, to validate the generalizability across different scales and pre-training paradigms, we have conducted comprehensive experiments on FARE-trained CLIP ViT-B/32 in **Appendix C.7 (Table A9)**, TeCoA-trained CLIP ViT-L/14 in **Appendix C.8 (Table A10)**, EVA-CLIP in **Appendix C.8 (Table A11)**.
>
> The results demonstrate that DEFEAT consistently enhances the adversarial robustness across different model scales and families, confirming that our framework is not limited to the specific CLIP.
>
> 2. **Generalization to other tasks**.
>
> In the adversarial prompt tuning domain, focusing on zero-shot/few-shot classification tasks has been the established common practice. To ensure a fair comparison, our evaluation pipeline **strictly aligns with the series of prior works**, including TeCoA [5, ICLR 2023], AdvPT [6, ECCV 2024], APT [7, CVPR 2024], and FAP [8, NeurIPS 2024], as well as recent related works TTC [2, CVPR2025], TAPT [3, CVPR2025], and R-TPT [4, CVPR2025].
>
> Furthermore, our evaluation is performed on **15 diverse benchmark datasets**, covering a wide spectrum of visual understanding tasks, including generic object classification, fine-grained visual categorization, texture classification, scene recognition, and action recognition. We believe this provides a robust and stable evaluation of our method's capabilities within this established research line.
>
> Nevertheless, extending DEFEAT to other applications is a meaningful idea. The focus of this paper is adversarial prompt tuning. We will investigate this direction in our future work.

---

> ### Author Response · Authors · 2025-11-22
> **Response by Authors, Part 2**
>
> > W3. The practical utility of the DEFEAT method is not fully clear due to the lack of analysis on its computational cost relative to baseline methods.
>
> ### Answer to W3:
>
> Thanks for your feedback. **We would like to clarify that a detailed computational cost analysis was indeed provided in Appendix C.10 (Table A13) as noted in the original submission (Lines 472-474), which may have been overlooked.** We summarize the key comparisons against baselines (APT and FAP) below:
>
> * Training Memory: DEFEAT's memory usage (21240M) is compareble with that of APT (21204M) and significantly lower than that of FAP (37500M).
> * Training Time: DEFEAT's training time (19.21h) is almost comparable with that of APT (18.99h) and less than half of that of FAP (39.04h).
> * Inference Time: About the inference time, DEFEAT is comparable with that of APT and FAP (3.1431ms vs. 2.9959ms vs. 3.0020).
>
> In summary, the computational overhead introduced by our method is minimal during both training and inference. We believe that such a slight cost is a highly favorable trade-off for the substantial improvement in robustness that DEFEAT provides over both APT and FAP, showing the practicality and efficiency of our method.
>
> ---
>
> > W4. The discussion could be further enriched by considering recent advancements in test-time defenses. if the authors could investigate whether combining DEFEAT with pre-processing defenses [1] or test-time defenses [2,3] could lead to synergistic effects, potentially achieving a "1+1>2" outcome.
>
> ### Answer to W4:
>
> Thanks for your insightful comment. According to your suggestion, **we combine our DEFEAT method with test-time defense methods** [3, 4], which **leads to synergistic effects**. Specifically, we design a novel variant, **DEFEAT-T**, which bridges train-time prompt tuning and test-time defense.
>
> The core of DEFEAT-T is to pivot our method from an "active defender" (via adversarial training) to a "passive purifier" (via clean training), combined with an optimization-free test-time ensemble. The mechanism of DEFEAT-T is introduced as follows.
>
> * **Phase 1: Clean Training**
> We first train our DEFEAT framework (including the learnable prompt and our PerturbShield module) on a **standard (non-robust) backbone using only clean data**. Here, PerturbShield learns to reconstruct the clean feature distribution.
>
> * **Phase 2: Test-Time Ensemble Defense**
> At test time, for each incoming sample, we do not perform any per-sample backpropagation or optimization, which is the bottleneck for test-time prompt tuning methods [3, 4]. Instead, we execute a forward-pass ensemble defense. **First**, we generate $N$ augmented views of the test sample. **Second**, all $N$ views are fed through our model trained from clean data to get $N$ fused logits by Eq. (7). **Next**, to obtain the final prediction, one can directly average these $N$ fused logits. Here, to have better performance, we perform a reliability-based weighted average. Specifically, we use the $N$ class features (which are collected alongside the grid features) to compute a similarity matrix. We then average the top-$k$ values in each row to get a reliability score for each augmented view. **Finally**, we use these reliability scores as weights for a weighted average of the corresponding fused logits to obtain the final prediction.
>
> The rationale for this design is our hypothesis that this view augmentation and weighted-average prediction could mitigate the impact of unseen adversarial perturbations, and that combining it with our DEFEAT framework provides a second stage of purification.

---

> ### Author Response · Authors · 2025-11-22
> **Response by Authors, Part 3**
>
> **Empirical Verification.**
>
> Our experiments confirm this "1+1>2" synergistic effect. As shown in the following table, DEFEAT-T (84.53% Rob.) significantly outperforms test-time prompt tuning methods like R-TPT (63.37% Rob.) and TAPT (56.38% Rob.), as well as our original DEFEAT (76.28% Rob.). For the results under the stronger $\epsilon=4/255$ attack, DEFEAT-T still outperforms baseline methods as well as DEFEAT, and we kindly refer the reviewer to our detailed **Answer to Q1** of Reviewer QZPj.
>
> To further isolate the contribution of our PerturbShield module and logits fusion strategy, we also extended the baseline APT to APT-T (i.e., APT trained on clean data, evaluated with the same test-time ensemble but without the logits fusion strategy and the PerturbShield module). While APT-T (73.03% Rob.) is strong, it is still significantly outperformed by DEFEAT-T (84.53% rob.). This comparison further demonstrates the effectiveness of our proposed logits fusion strategy and PerturbShield.
>
> |16shot,$\epsilon=1/255$|**Average**|||**Caltech101**||**OxfordPets**||**Flowers102**||**DTD**||
> |:---|:---:|:---:|:---:|:---:|:---:|:---:|:---:|:---:|:---:|:---:|:---:|
> |**Method**|**Acc.**|**Rob.**|**H**|**Acc.**|**Rob.**|**Acc.**|**Rob.**|**Acc.**|**Rob.**|**Acc.**|**Rob.**|
> |CLIP|72.38|3.43|6.54|91.03|11.52|87.38|1.39|66.91|0.08|44.21|0.71|
> |TeCoA|61.38|48.76|54.35|85.92|75.21|79.48|63.26|48.15|32.68|31.97|23.88|
> |
> |TAPT-VLI (reported)|/|56.38|/|/|82.10|/|68.10|/|44.60|/|30.70|
> |R-TPT|70.23|63.37|66.62|90.83|86.13|85.12|75.28|62.28|53.63|42.67|38.42|
> |
> |APT|79.73|65.56|71.95|92.90|83.98|83.95|65.14|86.80|73.57|55.26|39.54|
> |**DEFEAT**|80.04|76.28|78.11|93.39|91.12|84.71|79.53|84.90|81.81|57.15|52.66|
> |
> |APT-T|86.73|73.03|79.29|95.01|87.38|90.27|70.84|**94.56**|82.54|67.08|51.36|
> |**DEFEAT-T**|**87.78**|**84.53**|**86.12**|**96.19**|**95.58**|**91.96**|**90.13**|94.52|**88.35**|**68.44**|**64.07**|
>
> **Efficiency Trade-Off.**
>
> Crucially, our DEFEAT-T paradigm has a decisive efficiency advantage over existing test-time prompt tuning methods [3, 4]. Those methods require per-sample optimization (batch size=1) involving backpropagation, making them slow. Our DEFEAT-T is an inference-only scheme in test time with no gradient calculations, allowing for batch parallelism.
>
> As shown in the following table (tested on Caltech101), DEFEAT-T is ~35x faster than R-TPT (0.19s vs 6.63s) while achieving higher robustness. Compared to the original DEFEAT, it incurs a 2x inference cost but eliminates the need for adversarial training.
>
> In summary, DEFEAT and DEFEAT-T are suited for different scenarios. DEFEAT is ideal for low-latency inference, while DEFEAT-T is a powerful, highly robust, and computationally viable alternative when a robust backbone is not available. **We have included the above discussions in Appendix C.12 (highlighted in blue).**
>
> |Method|Train.Time (s)|Infer.Memory|Infer.Time (s/per img)|Acc.| Rob.| H|
> |:---|:---:|:---:|:---:|:---:|:---:|:---:|
> |**DEFEAT**|3597|**8481M**|**0.09**|93.39|91.12|92.24|
> |**DEFEAT-T**|**1079**|10420M|0.19|**96.19**|**95.58**|**95.88**|
> |R-TPT|0|5248M (batch=1)|6.63|90.83|86.13|88.42|
>
>
> [1] Diffusion Models for Adversarial Purification, ICML 2022
>
> [2] CLIP is Strong Enough to Fight Back: Test-time Counterattacks towards Zero-shot Adversarial Robustness of CLIP, CVPR 2025.
>
> [3] TAPT: Test-Time Adversarial Prompt Tuning for Robust Inference in Vision-Language Models, CVPR 2025.
>
> [4] R-TPT: Improving Adversarial Robustness of Vision-Language Models through Test-Time Prompt Tuning, CVPR 2025.
>
> [5] Understading Zero-shot Adversarial Robustness for Large-scale Models, ICLR 2023.
>
> [6] Adversarial Prompt Tuning for Vision-Language Models, ECCV 2024.
>
> [7] One PromptWord is Enough to Boost Adversarial Robustness for Pre-trained Vision-Language Models, CVPR 2024.
>
> [8] Few-Shot Adversarial Prompt Learning on Vision-Language Models, NeurIPS 2024.
>
> [9] Robust CLIP: Unsupervised Adversarial Fine-Tuning of Vision Embeddings for Robust Large Vision-Language Models, ICML 2024.

---

### Official Review · Reviewer_spep · 2025-11-01

**Soundness:** 2
**Presentation:** 3
**Contribution:** 3
**Rating:** 6
**Confidence:** 3

**Summary:**

This paper proposes DEFEAT, a robust prompt tuning framework for vision-language models (VLMs) that enhances adversarial robustness by leveraging discrete latent features. The method employs a VQ-VAE-based Perturbation Discrete Shield (PerturbShield) to map both clean and adversarial features into shared discrete representations, combined with logits fusion and prompt alignment strategies. Experiments across multiple datasets demonstrate significant improvements in adversarial robustness while maintaining competitive clean accuracy.

**Strengths:**

- The method is novel, introducing discrete latent representations into the robust prompt tuning framework, which provides a fresh and intuitive perspective on defending against adversarial perturbations.
- The experimental results are strong, showing substantial improvements in robustness accuracy compared with existing adversarial prompt tuning methods.
- The paper provides rich visualizations of feature distributions, which help illustrate and explain how the proposed method mitigates adversarial effects.

**Weaknesses:**

- The paper lacks a theoretical justification for why clean and adversarially perturbed features can be mapped to the same discrete representation within the VQ-VAE framework.
- The approach involves a large number of hyperparameters (\alpha, \beta, \lambda, \mu), making the method relatively hard to tune and potentially sensitive to configuration choices.

**Questions:**

Please refer to the weaknesses.

---

> ### Author Response · Authors · 2025-11-22
> **Response by Authors**
>
> Thank you for your constructive comments. Below we have made responses to your comments. If you have any further comment, please feel free to let us know and we are more than glad to discuss with you.
>
> ---
>
> > W1. The paper lacks a theoretical justification for why clean and adversarially perturbed features can be mapped to the same discrete representation within the VQ-VAE framework.
>
> ### Answer to W1:
>
> Thanks for your insightful suggestion. We provide a theoretical justification below.
> 1. **Theoretical Justification.**
>
> Let $\text{Enc}(\cdot)$ denote the composite visual encoder, which consists of the CLIP image encoder followed by the trainable VQ-VAE encoder. The **clean latent features** output by this composite encoder are denoted by $z_e^c = \text{Enc}(x) \in \mathbb{R}^{n \times d}$ (where $n$ is the number of latent positions after encoding, $d$ is the dimensionality of the latent embedding). Under an adversarial attack with perturbation $\delta$, the **adversarial latent features** become:
> $$
> z_e^a = \text{Enc}(x + \delta) = z_e^c + \Delta z,
> $$
> where $\Delta z \in \mathbb{R}^{n \times d}$ is the mapped perturbation in the latent space.
>
> **VQ-VAE Codebook and Covering Radius.** The core of VQ-VAE is to discretize continuous latent features through a **codebook** $\mathcal{E} = \\{e_k\\}_{k=1}^K$ (where $K$ is the codebook size). The **covering radius** $r$ of the codebook is defined as
>
> $$
> r = \max_{z \in \mathcal{Z}} \min_{e_k \in \mathcal{E}} \|z - e_k\|_2,
> $$
>
> where $\mathcal{Z}$ is the set of latent features output by the encoder from all clean images. The covering radius $r$ characterizes the "coverage capability" of the codebook over the clean feature space: the distance from any clean feature to its nearest codebook element does not exceed $r$, and $r$ is minimized through the codebook loss of VQ-VAE.
>
> Based on the above definition, if the perturbation magnitude in the latent space satisfies $\|\Delta z\|_2 \leq r$, it is statistically highly probable that both the clean feature $z_e^c$ and the adversarial feature $z_e^a$ map to the same codebook element $e_k \in \mathcal{E}$. Notably, the training objective in DEFEAT encourages images that are more similar at the pixel level to be closer in feature space. So the feature shift $\Delta z$ induced by minor perturbations is typically much smaller than the codebook covering radius $r$, effectively mitigating such adversarial attacks during the quantization process.
>
> 2. **Empirical Verification.**
>
> To validate this, we conduct a statistical analysis on the latent space using the Caltech101 dataset (16-shot, $\epsilon=4/255$). We measure two key metrics across all image patches:
>     * The average magnitude of adversarial perturbation per patch: $\|\Delta z\|_2 \approx \mathbf{126.0}$.
>     * The average distance between clean latent features and their assigned codebook vectors (serving as a proxy for the safety radius): $r \approx \mathbf{432.3}$.
>
> The experimental results show that the covering radius is approximately **3.43 times larger** than the perturbation magnitude ($432.3$ vs. $126.0$). This substantial margin implies that the vast majority of adversarial perturbations are insufficient to push the latent features out of their original discrete region defined by their assigned code. Consequently, the clean and adversarial patches are quantized to the same discrete codes, confirming the effectiveness of our defense. **We have incorporated this analysis and experimental result into Appendix C.13 (highlighted in blue).**
>
> ---
>
> > W2. The approach involves a large number of hyperparameters (\alpha, \beta, \lambda, \mu), making the method relatively hard to tune and potentially sensitive to configuration choices.
>
> ### Answer to W2:
>
> Thanks for your feedback. We address your concern as follows:
>
> 1. **Parameter Insensitivity**. As shown in **Fig. 5 (Sec. 4.3)** and **Fig. A8 (Appendix C.2)**, DEFEAT exhibits remarkably stable performance across a broad range of values for these parameters, making setting those hyperparameters very easy.
> 2. **Superiority Over Baselines**. Critically, the figures clearly show that simply enabling these components (i.e., setting those hyperparameter values $> 0$) is sufficient for DEFEAT to significantly outperform APT (20.26% Rob.) and FAP (19.82% in Rob.). Furthermore, DEFEAT maintains this superiority consistently across the entire tested range of hyperparameter values.
> 3. **Unified Configuration**. The strongest evidence against tuning difficulty is our experimental setup: as detailed in Sec. 4.1, **a single, fixed configuration ($\alpha=0.5, \beta=0.1, \lambda=10, \mu=20$) was used for all experiments across all 15 datasets.** The ability of one configuration to achieve state-of-the-art results on diverse benchmarks without dataset-specific tuning confirms the method's generalizability and ease of use.

---

### Official Review · Reviewer_QZPj · 2025-11-01

**Soundness:** 3
**Presentation:** 3
**Contribution:** 3
**Rating:** 6
**Confidence:** 4

**Summary:**

The authors propose DEFEAT (Discrete LatEnt Feature based Adversarial Training), a new robust prompt tuning framework for VLMs. The core hypothesis is that discretizing latent features can effectively mitigate adversarial perturbations. The authors conduct extensive experiments on 15 datasets across adversarial few-shot classification, domain generalization, and cross-dataset generalization settings. The results show that DEFEAT achieves state-of-the-art robustness and a better accuracy-robustness trade-off compared to existing adversarial prompt tuning methods like APT and FAP.

**Strengths:**

1. A Novel and Insightful Defense Mechanism: The core idea of using a VQ-VAE to create discrete latent feature representations as an adversarial defense is a novel contribution in the prompt-tuning space. The design is particularly insightful for choosing to discretize the grid features ($I_{patch}$) rather than the single, global class token ($I$).

2. Comprehensive experiments: Extensive experiments across 15 datasets are conducted to verify the effectiveness of the DEFEAT method. The paper provides good ablation studies that clearly demonstrate why the proposed components are necessary.

**Weaknesses:**

Heavy Reliance on a Pre-Trained Robust Backbone:  As the authors report in Section C, DEFEAT requires a robust visual backbone (like TeCoA) to function. When applied to a standard, non-robust CLIP model, the PGD attack can still pose significant threats.

**Questions:**

Since all tested adversarial prompt tuning methods fail without a robust backbone, is there any possible way to adapt your method under this scenario?

---

> ### Author Response · Authors · 2025-11-22
> **Response by Authors, Part 1**
>
> Thank you for your constructive comments. Below we have made responses to your comments. If you have any further comment, please feel free to let us know and we are more than glad to discuss with you.
>
> ---
>
> > W1. Heavy Reliance on a Pre-Trained Robust Backbone: As the authors report in Section C, DEFEAT requires a robust visual backbone (like TeCoA) to function. When applied to a standard, non-robust CLIP model, the PGD attack can still pose significant threats.
>
> ### Answer to W1:
>
> Thanks for your feedback. We would like to clarify this reliance and the contributions of our work as follows.
>
> * **This is a common constraint of existing methods**. As analyzed in Appendix C.9, **this reliance is a common constraint for current adversarial prompt tuning methods** (e.g., APT, FAP, and DEFEAT) under strong attack settings (e.g., $\epsilon=4/255$). As shown in our Table A12, DEFEAT performs well in a weaker attack setting (e.g., $\epsilon=1/255$), demonstrating significantly reduced the dependence on the TeCoA robust backbone.
>
> * **Our work is complementary to adversarial full fine-tuning.** We fully agree on the importance of a robust backbone. A significant body of excellent research is focusing on generating robust visual backbones via adversarial full fine-tuning (e.g., TeCoA [1], FARE [2], PMG-FT [3], AdvSimplex [4], and SLADE [5]). We believe that it is practical and beneficial to leverage these powerful, available backbones, and our work is designed to be complementary to, not in competition with, these efforts. Building on this complementary position, the advantage of DEFEAT is providing a mechanism to robustly adapt these powerful backbones to diverse downstream tasks more efficiently and effectively.
>
> * Furthermore, DEFEAT is a general framework, and its efficacy is not limited to a single robust visual backbone (i.e., TeCoA). **It successfully generalizes to other robust backbones like FARE, larger-scale models like ViT-L/14, and different VLM families like EVA-CLIP** (as shown in Tables A9, A10, and A11).
>
> In summary, the reliance on a robust backbone is a common constraint for current adversarial prompt tuning methods. DEFEAT's contribution is to provide a state-of-the-art, efficient, and robust framework for downstream adaptation within this constraint.
>
> Moreover, as detailed in the answer to the next question, we propose an extension, **DEFEAT-T**, which mitigates the heavy reliance by operating effectively without requiring a pre-trained robust backbone and even without adversarial training.
>
> ---
>
> > Q1. Since all tested adversarial prompt tuning methods fail without a robust backbone, is there any possible way to adapt your method under this scenario?
>
> ### Answer to Q1:
>
> Thanks for your insightful question. To address the scenario facing strong attacks without a pre-trained robust backbone, we propose **DEFEAT-T**, a variant of the DEFEAT method. DEFEAT-T **successfully adapts our core idea to the non-robust backbone scenario by combining it with a test-time defense strategy.**
>
> The core of DEFEAT-T is to pivot our method from an "active defender" (via adversarial training) to a "passive purifier" (via clean training), combined with an efficient, optimization-free test-time ensemble. The mechanism is structured into two phases:
>
> 1. **Phase 1: Clean Training**
> We first recognized that adversarial training on a non-robust backbone could be a source of the training collapse. Therefore, we **discard adversarial training**. Instead, we train our DEFEAT framework (including the learnable prompt and PerturbShield module) on a **standard (non-robust) backbone using clean data only**. In this phase, our PerturbShield module learns to reconstruct the clean feature distribution of the non-robust backbone.
>
> 2. **Phase 2: Test-Time Ensemble Defense**
> At test time, we execute forward-pass ensemble defense, avoiding per-sample backpropagation or optimization, which are major bottlenecks for test-time prompt tuning methods [6,7,8]. **First**, we generate $N$ (e.g., $N$=64) augmented views (e.g., random crop, random horizontal flip) of the test sample. **Second**, all $N$ views are fed through our model trained from clean data, to get $N$ fused logits by Eq. (7) as shown in Fig. 4. **Next**, to obtain the final prediction, one can directly average these $N$ fused logits. Here, to have better performance, we perform a reliability-based weighted average. Specifically, we use the $N$ class features (which are collected alongside the grid features) to compute a similarity matrix. We then average the top-$k$ (e.g., $k$=20) values in each row to get a reliability score for each augmented view. **Finally**, we use these reliability scores as weights for a weighted average of the corresponding fused logits to obtain the final prediction.

---

> ### Author Response · Authors · 2025-11-22
> **Response by Authors, Part 2**
>
> The rationale for this design is our hypothesis that this view augmentation and weighted-average prediction could mitigate the impact of unseen adversarial perturbations, and that combining it with our DEFEAT framework provides a second stage of purification.
>
> ### Empirical Verification.
>
> As shown in the table below, **DEFEAT-T (w/o TeCoA) achieves good average performance (87.76% Acc., 75.96% Rob., 81.43% H), surpassing the standard DEFEAT (w/ TeCoA) (68.77% Acc., 49.92% Rob. , 57.84% H) even without a robust backbone.**
>
> Simultaneously, we also extend the baseline APT to **APT-T (w/o TeCoA)** (i.e., APT trained on clean data, evaluated with the same test-time ensemble but without the logits fusion strategy and the PerturbShield module). We find that APT-T (w/o TeCoA) (43.33% Rob.) achieves performance comparable to the original DEFEAT (w/ TeCoA) (49.92% Rob.). However, its robustness is still significantly weaker than our DEFEAT-T (75.96% Rob.). This comparison further demonstrates the effectiveness of our proposed logits fusion strategy and PerturbShield.
>
> |16shot,$\epsilon=4/255$|**Average**|||**Caltech101**||**OxfordPets**||**Flowers102**||**DTD**||
> |:---|:---:|:---:|:---:|:---:|:---:|:---:|:---:|:---:|:---:|:---:|:---:|
> |**Method**|**Acc.**|**Rob.**|**H**|**Acc.**|**Rob.**|**Acc.**|**Rob.**|**Acc.**|**Rob.**|**Acc.**|**Rob.**|
> |CLIP|72.38|2.28|4.42|91.03|7.79|87.38|1.20|66.91|0.12|44.21|0.00|
> |TeCoA|49.87|19.86|28.41|78.78|43.61|66.26|15.56|30.13|8.93|24.29|11.35|
> |
> |APT(w/o TeCoA)|14.05|0.07|0.14|43.77|0.16|3.38|0.00|3.45|0.00|5.61|0.12|
> |DEFEAT(w/o TeCoA)|18.45|0.16|0.32|51.89|0.41|3.05|0.00|9.62|0.00|9.22|0.24|
> |
> |APT(w/ TeCoA)|68.96|33.94|45.49|86.29|56.75|67.29|19.98|76.41|37.52|45.86|21.51|
> |**DEFEAT**(w/ TeCoA)|68.77|49.92|57.84|87.63|78.30|72.17|30.77|71.13|58.26|44.15|32.33|
> |
> |APT-T(w/o TeCoA)|86.72|43.33|57.78|95.01|71.08|90.19|32.54|94.76|39.18|66.90|30.50|
> |**DEFEAT-T**(w/o TeCoA)|**87.76**|**75.96**|**81.43**|96.23|93.63|91.88|86.54|94.60|66.38|68.32|57.27|
>
> Crucially, our DEFEAT-T method has a decisive **efficiency advantage** over existing test-time prompt tuning methods [6,7,8]. Existing methods require per-sample optimization (batch size=1) involving backpropagation, making them slow. Our DEFEAT-T is an inference-only scheme with **no gradient calculations**, allowing for batch parallelism and much faster inference. However, even so, the time cost of DEFEAT-T's inference is higher than that of the original DEFEAT. As shown in the computational cost analysis shown in the following table (tested on Caltech101), the inference of **DEFEAT-T** is approximately **2x slower than that of DEFEAT**. On the other hand, the training time for DEFEAT-T is much shorter, as it does not require adversarial training.
>
> In summary, DEFEAT and DEFEAT-T can be adopted in different scenarios. The original DEFEAT (w/ TeCoA) is ideal for low-latency inference scenarios, whereas DEFEAT-T is a viable and highly robust alternative when a robust backbone is not available. **We have included the above discussions in Appendix C.12 (highlighted in blue)**.
>
> [1] Understading Zero-shot Adversarial Robustness for Large-scale Models, ICLR 2023.
>
> [2] Robust CLIP: Unsupervised Adversarial Fine-Tuning of Vision Embeddings for Robust Large Vision-Language Models, ICML 2024.
>
> [3] Pre-trained Model Guided Fine-tuning for Zero-shot Adversarial Robustness, CVPR 2024.
>
> [4] Improving Zero-Shot Adversarial Robustness in Vision-Language Models by Closed-form Alignment of Adversarial Path Simplices, ICML 2025.
>
> [5] SLADE: Shielding against Dual Exploits in Large Vision-Language Models, CVPR 2025.
>
> [6] Test-Time Prompt Tuning for Zero-Shot Generalization in Vision-Language Models, NeurIPS 2022.
>
> [7] TAPT: Test-Time Adversarial Prompt Tuning for Robust Inference in Vision-Language Models, CVPR 2025.
>
> [8] R-TPT: Improving Adversarial Robustness of Vision-Language Models through Test-Time Prompt Tuning, CVPR 2025.
>
> |Method|Train.Time (s)|Infer.Memory|Infer.Time (s/per img)|Acc.|Rob.|H|
> |:---|:---:|:---:|:---:|:---:|:---:|:---:|
> |DEFEAT (w/ TeCoA)|3597|**8481M**|**0.09**|87.63|78.30|82.70|
> |DEFEAT-T (w/o TeCoA)|**1079**|10420M|0.19|**96.23**|**93.63**|**94.91**|

---

### Author Response · Authors · 2025-11-29
**General Response**

Dear Reviewers and ACs:

We appreciate the diligent efforts and insightful comments from all reviewers and ACs. We are grateful the reviewers recognized our work for its novelty and insight in using latent feature discretization (Reviewers QZPj, spep, vWTM) and its comprehensive evaluation (all reviewers).

During the rebuttal, we made great efforts to address all concerns. Our major revisions are summarized in two parts:

(1) New Experiments:
* We added $\ell_2$ threat model validation. (**Table A14 of Appendix C.11**).
* We developed DEFEAT-T, a variant combining test-time defense, which eliminates the reliance on robust backbones and achieves SOTA performance without adversarial training. (**Table A15 and Table A16 of Appendix C.12**)
* We provided a theoretical justification analyzing the codebook covering radius versus perturbation magnitude to explain the defense mechanism, supported by empirical verification. (**Appendix C.13**)
* We expanded the evaluation of AutoAttack and CW attacks from 3 to 10 datasets. (**Table A7 of Appendix C.4**)
* We supplemented our results with multi-seed to ensure reproducibility. (**Table A17 and Table A18 of Appendix C.14**)
* We added experiments verifying result stability against prompt variations. (**Table A19 of Appendix C.15**)

(2) Clarifications:
* For Reviewer **QZPj**, we addressed the concern regarding the reliance on robust backbones by introducing the DEFEAT-T solution (**Appendix C.12**).
* For Reviewer **spep**, we addressed the request for theoretical justification (**Appendix C.13**) and clarified hyperparameter stability.
* For Reviewer **vWTM**, we respectfully clarified that the analysis of computational costs (Appendix C.10) and applicability to other VLM families (Appendix C.8) were already included in the original submission but might have been overlooked. Additionally, we addressed concerns regarding framework's integration, the scope of our evaluation tasks, and integration with test-time defenses (**Appendix C.12**).
* For Reviewer **DTmv**, we addressed questions regarding the validity of adaptive attack, failure cases, AA configs (**Appendix C.4**), codebook specs (**Appendix B**).

We believe our rebuttal has thoroughly addressed the reviewers' concerns and significantly improved the manuscript. Thank you for your time.

Best regards,

Authors of #10526

---

### Meta-Review · Area_Chair_hD2L · 2026-01-08

**Summary:**

This paper proposes DEFEAT, a robust prompt-tuning framework for VLMs based on discrete latent feature reconstruction, offering a clear and well-motivated alternative to costly adversarial fine-tuning while addressing a key vulnerability of continuous feature representations.

**Reviewer Concerns:**

This paper proposes DEFEAT, a robust prompt-tuning framework for VLMs based on discrete latent feature reconstruction, offering a clear and well-motivated alternative to costly adversarial fine-tuning while addressing a key vulnerability of continuous feature representations.

**Reviewer Scores:**

Most reviewers rated the paper at or above the acceptance threshold (typically 6), acknowledged the novelty and effectiveness of the discrete latent defense, and indicated that their main concerns were resolved after the rebuttal; no critical blocking issues remain.

Given vWTM’s concerns(the only negative review) were directly addressed with concrete additions—non-CLIP backbones already evaluated (EVA-CLIP, ViT-L/14, FARE-trained CLIP), explicit compute tables, and a new DEFEAT-T test-time variant with strong “1+1>2” results—they would likely have moved from 4 to 6 (at least “marginally above threshold”).

---

### Decision · Program_Chairs · 2026-01-26

Accept (Poster)